# Cholecystokinin facilitates motor skill learning by modulating neuroplasticity in the motor cortex

Hao Li[1,2], Jingyu Feng[1], Mengying Chen[1], Min Xin[1,2], Xi Chen[1], Wenhao Liu[1], Liping Wang[3], Kuan Hong Wang[4]*, Jufang He[1,2]*

[1]Departments of Neuroscience and Biomedical Sciences, City University of Hong Kong, Hong Kong, China; [2]Centre for Regenerative Medicine and Health, Hong Kong Institute of Science & Innovation, Chinese Academy of Sciences, Hong Kong, China; [3]The Brain Cognition and Brain Disease Institute (BCBDI), Shenzhen Institutes of Advanced Technology, Chinese Academy of Sciences, Shenzhen, China; [4]Department of Neuroscience, Del Monte Institute for Neuroscience, University of Rochester Medical Center, Rochester, United States

*For correspondence:
kuanhong_wang@urmc.rochester.edu (KHongW);
jufanghe@cityu.edu.hk (JH)

Competing interest: The authors declare that no competing interests exist.

**Abstract** Cholecystokinin (CCK) is an essential modulator for neuroplasticity in sensory and emotional domains. Here, we investigated the role of CCK in motor learning using a single pellet reaching task in mice. Mice with a knockout of *Cck* gene (*Cck*$^{-/-}$) or blockade of CCK-B receptor (CCKBR) showed defective motor learning ability; the success rate of retrieving reward remained at the baseline level compared to the wildtype mice with significantly increased success rate. We observed no long-term potentiation upon high-frequency stimulation in the motor cortex of *Cck*$^{-/-}$ mice, indicating a possible association between motor learning deficiency and neuroplasticity in the motor cortex. In vivo calcium imaging demonstrated that the deficiency of CCK signaling disrupted the refinement of population neuronal activity in the motor cortex during motor skill training. Anatomical tracing revealed direct projections from CCK-expressing neurons in the rhinal cortex to the motor cortex. Inactivation of the CCK neurons in the rhinal cortex that project to the motor cortex bilaterally using chemogenetic methods significantly suppressed motor learning, and intraperitoneal application of CCK4, a tetrapeptide CCK agonist, rescued the motor learning deficits of *Cck*$^{-/-}$ mice. In summary, our results suggest that CCK, which could be provided from the rhinal cortex, may surpport motor skill learning by modulating neuroplasticity in the motor cortex.

## Editor's evaluation

This important study investigates the contribution of Cholecystokinin (CCK), a neurotransmitter known to be involved in sensory and emotional function, to motor learning. The authors provide convincing evidence combining behavioural assays, brain recording and stimulation experiments, knock out models, and targeted manipulations of several relevant pathways that support a contribution of rhinal CCK projections to motor cortex during learning in mice. This paper thus identifies a potential novel pathway involved in motor learning, but the specific contribution of rhinal CCK still needs to be fully characterised in future work given the extensive projections of rhinal CCK neurons to brain areas other than motor cortex.

## Introduction

Learning to perform motor skills is essential for survival and high quality of life, such as hunting, running, escaping, fighting, playing music, dancing, drawing, and performing an operation. Evidence from electrical stimulation, lesions, imaging, and more targeted manipulation shows that the motor cortex is the center that controls motor behaviors and motor skill learning in the brain (*Papale and Hooks, 2018*). Changes among neuronal circuits, such as synaptic strength, circuit connectivity, neuronal excitability, and neuronal structure, which occur through all layers of the motor cortex, contribute to motor skills learning (*Papale and Hooks, 2018*; *Biane et al., 2016*; *Peters et al., 2014*; *Costa et al., 2004*; *Huber et al., 2012*). Different layers exhibit various neuronal changes with motor skill learning, corresponding with various layer-specific inputs and descending outputs. However, it is not completely clear how neuroplasticity in the motor cortex is regulated.

Cholecystokinin (CCK), distributed throughout the whole brain, has been suggested to be important in neuroplasticity (*Li et al., 2014*; *Chen et al., 2019*). Activation of the CCK-B receptor (CCKBR) by infusion of agonist in the auditory cortex regulated visuo-auditory associative memory formation in awake rats (*Li et al., 2014*). Projections from the entorhinal cortex of the medial temporal lobe release CCK in the neocortex, hippocampus, and amygdala, enabling the encoding of long-term associative, spatial, and fear memory (*Li et al., 2014*; *Meunier et al., 1996*; *Chen et al., 2019*; *Su et al., 2019*; *Feng et al., 2021*). *N*-Methyl-D-aspartate (NMDA) receptors in the presynaptic membrane control the release of the entorhinal CCK in the auditory cortex (*Chen et al., 2019*).

Motor memory, known as a subset of procedural memory, is quite distinct from declarative memory examined in previous studies of CCK functions, but both types involve neuronal changes in the neocortex caused by task training (*Squire, 2004*; *Ackermann and Rasch, 2014*). In this study, we examined the role of CCK from the rhinal cortex, including the entorhinal cortex and perirhinal cortex, to the motor cortex in neuroplasticity and motor skill learning. We evaluated whether the motor learning ability of mice is affected by the genetic elimination of the *Cck* gene or administration of the CCKBR antagonist. We implemented calcium imaging of the motor cortex to establish whether the absence of CCK function disrupts the refinement of the neuronal activation pattern during motor training. We further examined immunohistochemically the CCK-positive neuronal projections from the rhinal cortex to the motor cortex, including the laminar specificity of these projections in their target regions. In the final set of behavioral studies, we investigated whether the loss-of-function by inactivating CCK neurons from the rhinal cortex to the motor cortex suppresses motor learning ability and the gain-of-function by CCK4 administration rescues the motor learning ability of $Cck^{-/-}$ mice.

## Results

### The role of CCK in motor learning

A previous study has demonstrated that CCK is a key factor regulating neuroplasticity that enhances long-term memory formation in the auditory cortex (*Chen et al., 2019*). Therefore, we introduced the single pellet reaching task to train transgenic $Cck^{-/-}$ mice and their wildtype (WT) control (C57BL/6) to use the dominant forelimbs and obtain food rewards, as the method to determine whether CCK is involved in motor learning (*Figure 1A*). This task, including shaping and training, has been implemented in numerous studies on motor skill learning and motor control systems, especially those related to the forelimb movement (*Figure 1B*; *Xu et al., 2009*; *Wang et al., 2017*). The performance of both WT and $Cck^{-/-}$ mice was evaluated based on the success rate in the task that requires accurate performance in aiming, reaching, grasping, and retrieving (*Video 1*). The success rate of $Cck^{-/-}$ mice did not increase after 6 days of training, remaining at the baseline level of approximately 15% (*Figure 1C*, *Figure 1—figure supplement 1A, B*; $Cck^{-/-}$ mice, one-way repeated measures (RM) analysis of variance [ANOVA], $F[5,35] = 0.574$; $p = 0.72$; post hoc pairwise comparison between different days, Day 1 vs. Day 3, 15.05% ± 4.40% vs. 11.91% ± 3.60%, $p = 0.59$; Day 1 vs. Day 6, 15.05% ± 4.40% vs. 15.59% ± 3.36%, $p = 0.924$), while WT mice performed much better, of which the success rate increased significantly to 30.94% on Day 3 and remained at a plateau until the end of training (*Figure 1C*; WT mice, one-way RM ANOVA, $F[5,45] = 4.904$; $p < 0.001$; post hoc pairwise comparison, Day 1 vs. Day 3, 14.63% ± 3.05% vs. 30.94% ± 4.17%, $p = 0.013 < 0.05$; Day 1 vs. Day 6, 14.6% ± 3.05% vs. 32.76% ± 3.12%, $p = 0.004 < 0.01$; between WT and $Cck^{-/-}$ mice, two-way mixed ANOVA, significant interaction, $F[5,80] = 4.03$, $p = 0.003 < 0.01$; post hoc comparison bewteen two groups,

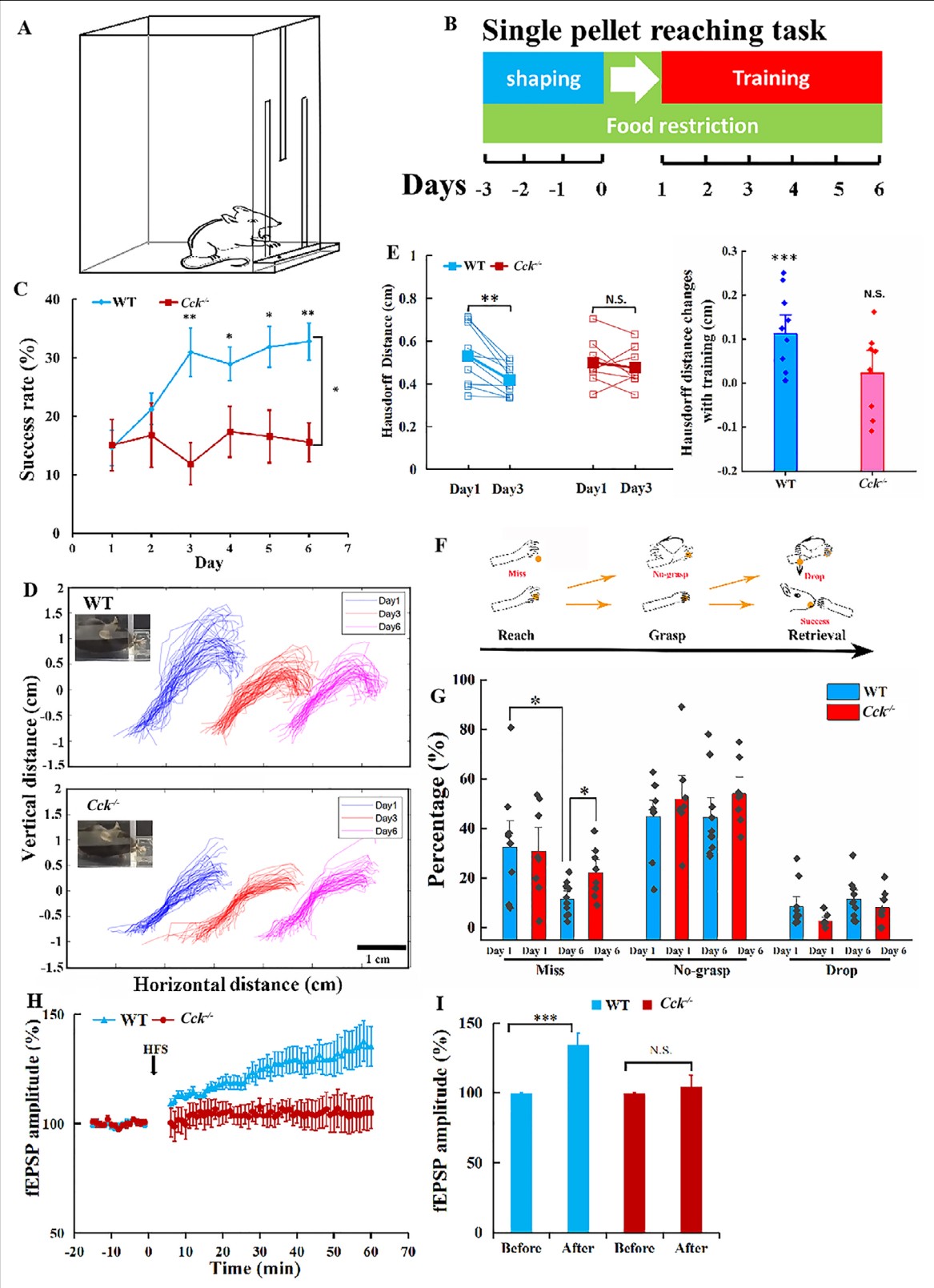

**Figure 1.** Single pellet reaching task for *Cck*⁻/⁻ and WT mice. (**A**) Task schematic. A mouse reaches for the food pellet through the slit. (**B**) Procedure. Three days before training, the mouse was placed in the chamber and allowed to acclimate to the environment and determine the dominant hand. Throughout the procedure, the mouse was food restricted, keeping the body weight at approximately 90% of the original weight. (**C**) Success rate of wildtype (WT, C57BL/6) (*N* = 10) and *Cck*⁻/⁻ (*N* = 8) mice performing the single pellet reaching task. *p < 0.05, **p < 0.01. Two-way mixed analysis of

*Figure 1 continued on next page*

*Figure 1 continued*

variance (ANOVA), post hoc comparison between two groups. (**D**) Representative trajectories of WT and *Cck*⁻/⁻ mice at Days 1, 3, and 6. (**E**) The pairwise Hausdorff distances of the trajectories were calculated to compare the variation in the trajectories of WT (*N* = 10) and Cck⁻/⁻ mice (*N* = 8). Left, blue and red solid square represent for average of the Hausdorff distance of WT and *Cck*⁻/⁻ mice, respectively. **p < 0.01, N.S. means not significant. Paired *t*-test. Right, Hausdorff distance changes with 3-day training of WT and *Cck*⁻/⁻ mice. ***p < 0.001, N.S. means not significant. *t*-test. (**F**) Diagram shows the task phases (reach, grasp, and retrieval) and different reaching results (miss, no-grasp, drop, and success). (**G**) Detailed reaching results for WT and Cck⁻/⁻ mice on experimental Days 1 and 6. *p < 0.05; paired *t*-test and *t*-test. (**H**) Normalized field excitatory postsynaptic potential (EPSP) amplitude before and after high-frequency stimulation (HFS) for both WT (*N* = 6, *n* = 21) and *Cck*⁻/⁻ mice (*N* = 3, *n* = 7). (**I**) The average normalized field excitatory postsynaptic potential (fEPSP) amplitude 10 min before HFS (−10 to 0 min, before) and 10 min after HFS (50 to 60 min, after) in the two groups of mice. ***p < 0.001, N.S. means not significant. Two-way mixed ANOVA, pairwise comparison.

The online version of this article includes the following figure supplement(s) for figure 1:

**Figure supplement 1.** Learning curve of single mouse of *Cck*⁻/⁻ (**A**) and wildtype (**B**) group and basic movement ability of WT and *Cck*⁻/⁻, including stride length (**C**, *t*-test, p = 0.405), stride time (**D**, *t*-test, p = 0.973), step cycle ratio (**E**, *t*-test, p = 0.093), and grasp force (**F**, *t*-test, p = 0.543).

$F_{[1,16]}$ = 7.697, p = 0.014 < 0.05; WT vs. *Cck*⁻/⁻, Day 3, 30.94% ± 4.17% vs. 11.91% ± 3.60%, $F_{[1,16]}$ = 11.239, p = 0.004 < 0.01; Day 4, 28.96% ± 2.90% vs. 17.37% ± 4.35%, $F_{[1,16]}$ = 5.266, p = 0.036 < 0.05; Day 5, 31.90% ± 3.50% vs. 16.56% ± 4.51%, $F_{[1,16]}$ = 7.465, p = 0.015 < 0.05; Day 6, 32.76% ± 3.12% vs. 15.59% ± 3.36%, $F_{[1,16]}$ = 13.906, p = 0.0018 < 0.01). The success rates of WT and *Cck*⁻/⁻ mice were similar on Day 1, indicating that CCK did not affect the basic ability to carry out the task, although the learning ability was inhibited (*Figure 1C*; *t*-test, WT vs. *Cck*⁻/⁻, 14.62% ± 3.05% vs. 15.05% ± 4.40%, p = 0.9366). Comparative analysis of the stride length, stride time, step cycle ratio, and grasp force of *Cck*⁻/⁻ mice with those of WT mice further established that CCK did not affect their basic movement ability (*Figure 1—figure supplement 1C–F–*). We also evaluated the variation of trajectories of the forelimb movement. The deviation of the trajectories of different trials of a WT mouse became visibly smaller on Day 3 compared with that on Day 1, while that of a *Cck*⁻/⁻ mouse showed no visible improvement (*Figure 1D*). We calculated the Hausdorff distances, the greatest of all the distances from a point in one set to the closest point in the other set, to evaluate the variation of trajectories (*Aydin et al., 2021*). The Hausdorff distance for the trajectories of WT and *Cck*⁻/⁻ mice are similar at Day 1 (*Figure 1E*; *t*-test, WT vs. *Cck*⁻/⁻, 0.53 ± 0.04 vs. 0.50 ± 0.04 cm, p = 0.5908). However, after 3 days' training, the Hausdorff distance for WT mice significantly decreased while *Cck*⁻/⁻ mice remained unchanged (*Figure 1E*; paired *t*-test, WT, Day 1 vs. Day 3, 0.53 ± 0.04 vs. 0.42 ± 0.02 cm, p = 0.003 < 0.01; *Cck*⁻/⁻, Day 1 vs. Day 3, 0.50 ± 0.04 vs. 0.48 ± 0.03 cm, p = 0.514).

Failures in retrieving the pellets, including miss, no-grasp, and dropping, are also applied to assess specific learning defects in different movement phases of the complex task, comprising the deficiency of 'success', which only indicates the final execution results (*Figure 1F*). 'Miss', representing no touching of the food pellet in front of the wall of the chamber, is due to inaccurate aiming and inadequate preparation of the neuronal system, especially processes involved in motor control and execution (*Video 2*). A 'no-grasp' is a reach attempt in which the mouse shows a defect in finger closure around food pellets for retrieval (*Video 3*). A 'drop' refers to a reach where the mouse drops the food pellet before putting it into the mouth, although the pellet was grasped correctly, indicating a defect in neurons controlling the retrieval process (*Video 4*). The miss rate of *Cck*⁻/⁻ mice was higher than that of WT mice, suggesting that CCK may affect the learning ability in aiming and preparing to execute a motor task (*Figure 1G*; paired *t*-test,

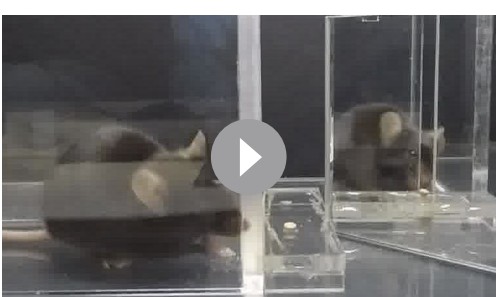

**Video 1.** Example of 'miss' performance of a mouse.
https://elifesciences.org/articles/83897/figures#video1

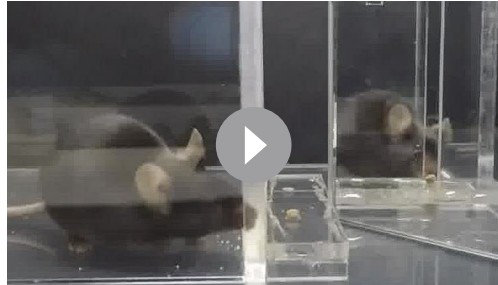

**Video 2.** Example of 'no-grasp' performance of a mouse.
https://elifesciences.org/articles/83897/figures#video2

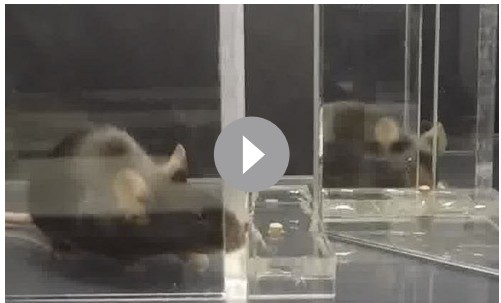

**Video 3.** Example of 'drop' performance of a mouse.
https://elifesciences.org/articles/83897/figures#video3

WT, Day 1 vs. Day 6, 32.54% ± 6.43% vs. 11.62% ± 3.58%, p = 0.013 < 0.05; $Cck^{-/-}$, Day 1 vs. Day 6, 30.77% ± 7.07% vs. 22.25% ± 2.09%, p = 0.173; t-test, WT vs. $Cck^{-/-}$, Day 1, 32.54% ± 6.43% vs. 30.77% ± 7.07%, p = 0.937, Day 6, 11.62% ± 3.58% vs. 22.25% ± 2.09%, p = 0.027 < 0.05). In contrast, in terms of no-grasp and drop, there were no significant changes after 6 days training for both $Cck^{-/-}$ and WT mice (*Figure 1G*; paired t-test, for 'no-grasp', WT, Day 1 vs. Day 6, 44.84% ± 4.43% vs. 44.53% ± 5.31%, p = 0.966; $Cck^{-/-}$, Day 1 vs. Day 6, 53.37% ± 4.86% vs. 54.02% ± 4.47%, p = 0.891; for 'drop', WT, Day 1 vs. Day 6, 8.52% ± 2.79% vs. 11.45% ± 2.55%, p = 0.418; $Cck^{-/-}$, Day 1 vs. Day 6, 4.94% ± 1.83% vs. 8.04% ± 2.46%, p = 0.254).

Furthermore, we conducted an electrophysiology experiment on the slices of the motor cortex from WT and $Cck^{-/-}$ mice to investigate the potential physiological causes for the defects in motor skill learning of $Cck^{-/-}$ mice. We observed long-term potentiation (LTP) in field excitatory postsynaptic potential (fEPSP) after high-frequency stimulation (HFS) in the WT mice, but no LTP in $Cck^{-/-}$ mice, suggesting that CCK plays a key role in neuroplasticity in the motor cortex (*Figure 1H, I*; two-way mixed ANOVA, $F_{[1,24]}$ = 3.154, p = 0.088; post hoc pairwise comparison, WT, before vs. after HFS, 100.06% ± 0.35% vs. 134.38% ± 8.61%, $F_{[1,20]}$ = 17.255, p < 0.001; $Cck^{-/-}$, before vs. after, 99.82% ± 0.48% vs. 104.62% ± 7.99%, $F_{[1,6]}$ = 0.5, p = 0.506; t-test, before HFS, WT vs. $Cck^{-/-}$, 100.06% ± 0.35% vs. 99.82% ± 0.48%, p = 0.787).

In summary, $Cck^{-/-}$ mice showed an impaired ability in motor skill learning in the single pellet reaching task and a defect in the LTP induction in the motor cortex.

## A CCKBR antagonist injection in the motor cortex inhibited the motor learning ability of C57BL/6 mice

As deletion of the *Cck* gene in the $Cck^{-/-}$ mouse is general, the above experiment results could not indicate the source of CCK and their action site in the brain. We limited our manipulation of the CCK signaling in the motor cortex, targeting its primary receptor, CCKBR, in the neocortex. We have implanted a drug infusion cannula into the motor cortex contralateral to its dominant forelimb and injected the CCKBR antagonist, L365.260 or its vehicle control to examine whether blocking the CCKBRs in the motor cortex could affect motor skill learning (*Figure 2A*, *Figure 2—figure supplement 1D*).

We infused L365.260 to the experimental group or vehicle (artificial cerebral spinal fluid [ACSF] + 0.1% dimethyl sulfoxide (DMSO)) to the control group through the implanted drug cannula in the motor cortex every day before training. The success rate of pellet retrieval of the experimental group was not improved through the 6-day training period (*Figure 2B*, *Figure 2—figure supplement 1A*; one-way RM ANOVA, $F_{[5,50]}$ = 1.959, p = 0.101), while that of the vehicle control group was significantly improved to 32.30% at Day 3 and kept at this level till the end of training (*Figure 2B*, *Figure 2—figure supplement 1B*; one-way RM ANOVA, pairwise comparison, Day 1 vs. Day 3, 19.02% ± 4.27% vs. 32.30% ± 3.62%, p = 0.038 < 0.05; Day 3 vs. Day 6, 32.30% ± 3.62% vs. 32.90% ± 7.07%, p = 0.937; Day 1 vs. Day 6, 19.02% ± 4.27% vs. 32.90% ± 7.07%, p = 0.064). The differences in the success rate between the experimental and control groups were significant (two-way mixed ANOVA, $F_{[5,70]}$ = 1.881, p = 0.109; post hoc comparison between Antagonist and Vehicle, $F_{[1,14]}$ = 5.066, p = 0.041; Day 3, Antagonist vs. Vehicle, 16.80% ± 2.83% vs. 32.30% ± 3.62%, $F_{[1,15]}$ = 11.266, p = 0.0048 < 0.01; Day 4,

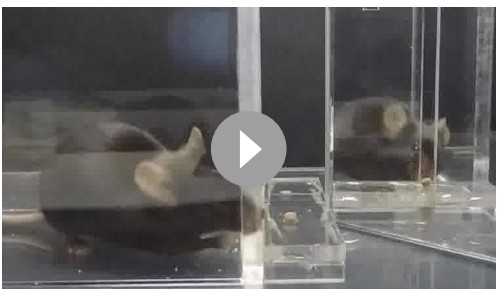

**Video 4.** Example of 'success' performance of a mouse.
https://elifesciences.org/articles/83897/figures#video4

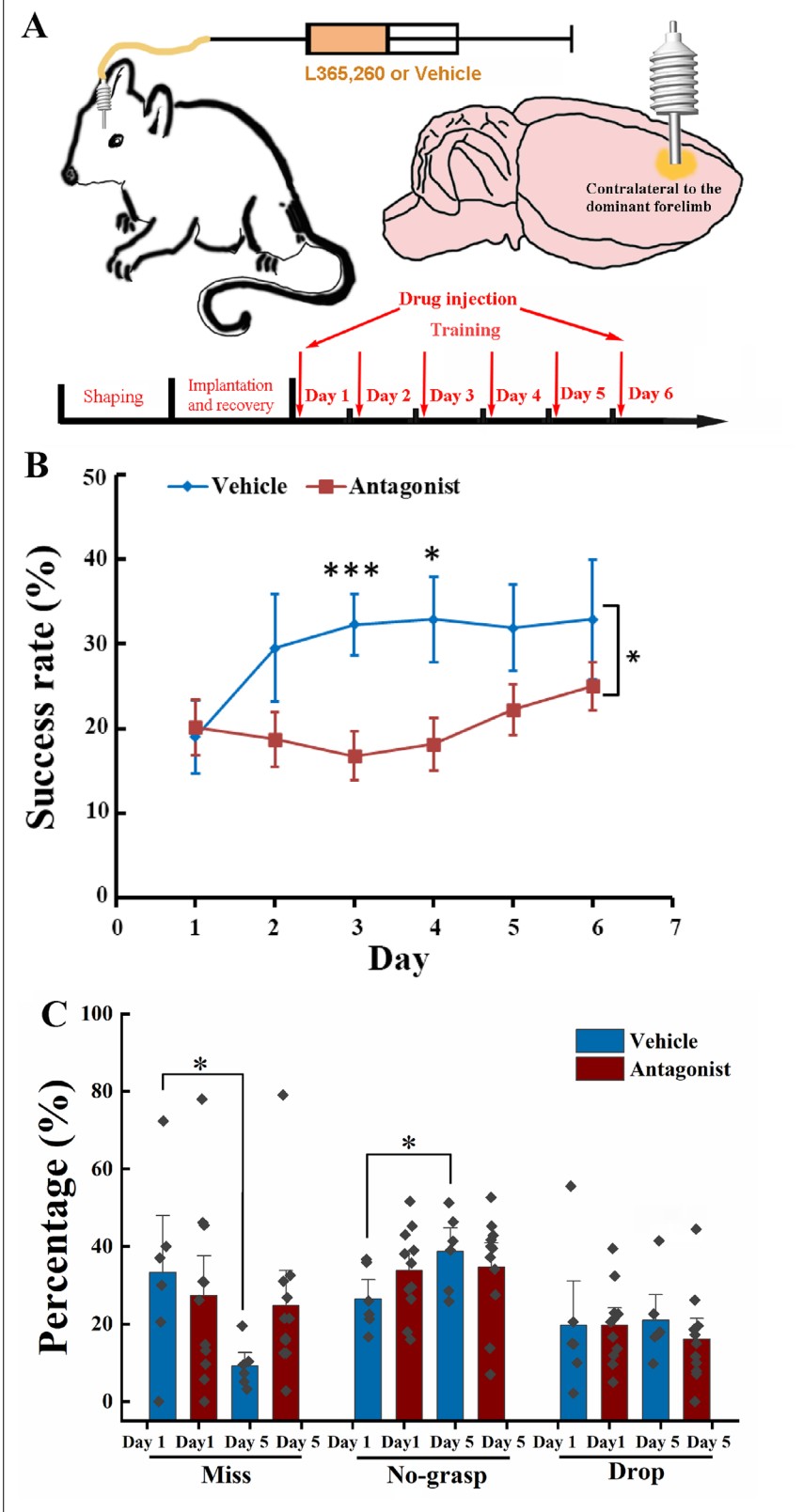

**Figure 2.** Effect of local injection of CCK-B receptor (CCKBR) antagonist on motor learning. (**A**) A cannula was implanted into the motor cortex contralateral to the dominant hand. One microliter of L365.260 or vehicle was injected into the motor cortex through the cannula every day before training. (**B**) Success rate of the mice injected with CCKBR antagonist ($N = 11$) and vehicle ($N = 6$). *$p < 0.05$, ***$p < 0.001$. Two-way mixed analysis of variance

*Figure 2 continued on next page*

*Figure 2 continued*

(ANOVA), post hoc comparison between two groups. (**C**) Detailed reaching results, in terms of miss, no-grasp, drop, on Days 1 and 5 for mice injected with CCKBR antagonist and vehicle. *p < 0.05, paired *t*-test.

The online version of this article includes the following figure supplement(s) for figure 2:

**Figure supplement 1.** Learning curve of every single mouse administrated with CCK-B receptor (CCKBR) Antagonist (**A**) and Vehicle (**B**) and example of drug cannula implantation site in the motor cortex (MC) (**C**) and detailed learning results of the learned mice before and after injected with antagonist (**D**), paired *t*-test, Before vs. after, Miss, p = 0.099, No-grasp, p = 0.506, Drop, p = 0.853, success, p = 0.854.

18.16% ± 3.12% vs. 32.90% ± 5.03%, $F[1,15]$ = 6.876, p = 0.019 < 0.05). This result suggests that CCK participates in motor skill learning by regulating neuroplasticity in the motor cortex.

For the detailed reaching results, we compared the performance of the experimental group and the control group on Days 1 and 5. The number of 'miss' of the Antagonist group had no significant decrease with learning, but for the Vehicle group, it dropped from 35% to 10%, indicating that the aiming and advance learning abilities were significantly impaired by inactivating CCKBRs in the motor cortex (*Figure 2C*, paired *t*-test, Antagonist, Day 1 vs. Day 5, 27.34% ± 9.85% vs. 24.75% ± 2.34%, p = 0.794; Vehicle, Day 1 vs. Day 5, 33.05% ± 6.68% vs. 9.17% ± 6.04%, p = 0.044 < 0.05). For the 'no-grasp' outcome, the Vehicle group increased significantly by 12.24%, indicating that the implantation of a cannula may cause injury to the motor cortex, leading to defects in digit learning (*Figure 2C*; paired *t*-test, 'no-grasp', Day 1 vs. Day 5, 26.49% ± 3.26% vs. 38.73% ± 4.05%, p = 0.017 < 0.05), while the experimental group showed no improvement (paired *t*-test, 'no-grasp', Day 1 vs. Day 5, 33.78% ± 3.36% vs. 34.69% ± 4.12%, p = 0.85). The drug cannula was implanted into the motor cortex by inserting the cannula below the brain surface by 250–300 μm, which inevitably caused injury to the motor cortex (*Figure 2—figure supplement 1C*). The inconsistent result between the experimental group and the control group is because there was no improvement from 'miss' to 'no-grasp' for the experimental group, leaving no change in the 'no-grasp' rate. The drop rate of both groups had no significant changes, indicating that the retrieval learning ability was not affected (*Figure 2C*; paired *t*-test, Vehicle, Day 1 vs. Day 5, 19.64% ± 3.01% vs. 16.15% ± 3.55%, p = 0.542; Antagonist, Day 1 vs. Day 5, 18.29% ± 6.57% vs. 22.71% ± 4.14%, p = 0.373). We also took Day 3 into the comparison with Days 1 and 5. For the Antagonist group, the detailed reaching result is comparable with that of Day 1 or 5. For the Vehicle group, though the miss rate of Day 3 is between that of Days 1 and 5, it did not significantly lower than that of Day 1, which could be because that 3 days' training is not enough for the miss rate to reach the plateau (*Figure 2—figure supplement 1E*). In summary, CCK plays a critical role in memory acquisition by activating the CCK receptors in the motor cortex at the overall level.

To exclude the possibility that the antagonist impaired the movement ability, we injected L365.260 into mice that had learned the single pellet reaching task. In terms of 'miss', 'no-grasp', 'drop', and 'success', there was no significant difference between before and after the injection of L365.260 (*Figure 2—figure supplement 1D*).

## Calcium imaging of layer 2/3 of the motor cortex during motor skill learning

Based on the outcome of the above drug infusion experiment and previous studies, the motor cortex is one of the primary sites for motor skill learning (*Wang et al., 2017*). Previous studies found that neuronal activity patterns in the layer 2/3 of the motor cortex were refined, exhibiting reproducible spatiotemporal sequences of activities with motor learning (*Peters et al., 2014*). Therefore, calcium imaging of neurons in the motor cortex layer 2/3 of C57BL/6 mice, *Cck*$^{-/-}$ mice, and C57BL/6 mice injected (i.p.) with the CCKBR antagonist was performed to determine the activity of neurons in the motor cortex during the single pellet reaching task.

We hypothesized that the CCK-enabled neuroplasticity happens at the population level in the motor cortex. To test the hypothesis, we attached a one-photon miniscope over the motor cortex, contralateral to the dominant hand of the mouse, with an implanted high light transmission glass window in between (*Figure 3B*). We installed a web camera in front of the training chamber to simultaneously monitor the mouse performing the task with the neuronal activity.

We recruited three groups of mice, (1) C57BL/6, (2) *Cck*$^{-/-}$, and (3) C57BL/6 with CCKBR antagonist, to examine how CCK signaling affects neuronal activity in the motor cortex (*Figure 3A*). We first

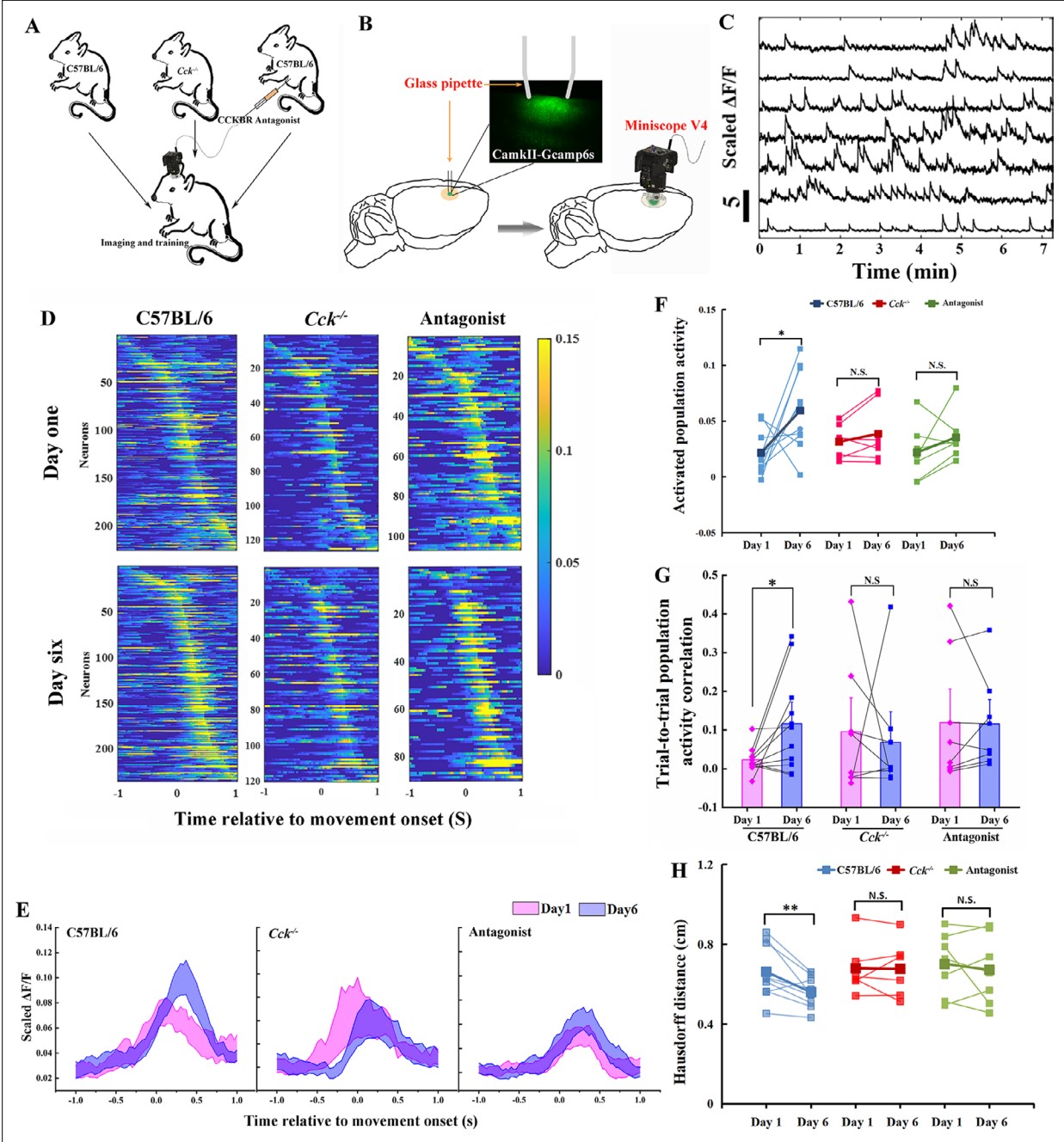

**Figure 3.** Calcium imaging of the MC during motor skill learning. (**A**) Experiment setup. C57BL/6, *Cck*$^{-/-}$, and C57BL/6 mice injected with CCK-B receptor (CCKBR) antagonist were applied for single pellet reaching task training and calcium imaging. (**B**) Schematic diagram of calcium imaging. A wide-tip glass pipette tightly touched the brain by being lowered to a depth of 400–500 µm, and strong GCaMP6s virus expression was observed in the superficial layer of the motor cortex with a high contrast compared with the deep layers after >14 days of expression. A baseplate was implanted on the skull, which was connected to the miniscope for calcium imaging during motor skills training (right panel). (**C**) Representative traces of extracted neurons from miniscope using the CNMF-E algorithm. The scale bar represents 5 units of the scaled $\Delta F/F$. (**D**) Neuronal activity pattern of C57BL/6 (N = 10), *Cck*$^{-/-}$ (N = 7), and C57BL/6 mice injected with L365.260 (N = 7). Upper line is from training Day 1 and the bottom is from training Day 6. (**E**) Neuronal population activity from C57BL/6, *Cck*$^{-/-}$, and C57BL/6 mice injected with L365.260. (**F**) Activated population activity (peak activity minus baseline activity) was calculated for C57BL/6, *Cck*$^{-/-}$, and C57BL/6 mice injected with L365.260 at Days 1 and 6. *p < 0.05, N.S. not significant. Paired *t*-test. (**G**) Trial-to-trial population activity correlation at Days 1 and 6 for C57BL/6, *Cck*$^{-/-}$, and C57BL/6 injected with L365.260. (**H**) The pairwise Hausdorff distances of the trajectories for C57BL/6, Cck$^{-/-}$, and C57BL/6 injected with L365.260 at Days 1 and 6. *p < 0.05, **p<0.01 N.S., not significant. One-way RM analysis of variance (ANOVA).

The online version of this article includes the following figure supplement(s) for figure 3:

*Figure 3 continued on next page*

*Figure 3 continued*

**Figure supplement 1.** Neuronal activity relative to the movement of different groups, including C57BL/6 (**A, B**), *Cck*⁻/⁻ (**C, D**), and L365,260 injection (**E, F**) mice at Days 1 and 6.

**Figure supplement 2.** Increase of trial-to-trial population activity correlation (C57BL/6, p = 0.01; *Cck*⁻/⁻, p = 0.61; Antagonist, p = 0.53).

confirmed GCaMP6s signals in layer 2/3 of the motor cortex, as shown in the examples (*Figure 3B*; *Video 5*). The neuronal signals were extracted with CNMF-E (*Zhou et al., 2018*) and analyzed with MATLAB (*Figure 3C*). Neurons showed various temporal and spatial responses to the movements during the task.

The neuronal activity pattern, excluding the indiscriminate neurons (ranksum test, neuronal activity during reaching and not reaching, p ≥ 0.05), in the C57BL6 group, was refined after 6 days of training; the peak activity of the neurons became stronger with lower background activity (*Figure 3D*). These results are similar to that of layer 2/3 neurons of the motor cortex in a mouse performing a lever-press task (*Peters et al., 2014*). In contrast, we found no apparent changes after training for 6 days for groups of *Cck*⁻/⁻ and C57BL/6 mice injected with the antagonist, in the neuronal activity pattern (*Figure 3D*).

The population activity of neurons varied with time relative to movement onset, starting to rise around 0.2 s before movement onset and reaching the peak at the time of 0.33 s after movement onset (*Figure 3E* and *Figure 3—figure supplement 1*). The activated population activity, peak activity minus baseline activity, for C57BL/6 mice increased significantly with training (*Figure 3F*; paired *t*-test, Day 1 vs. Day 6, 0.0216 ± 0.0062 vs. 0.0593 ± 0.0114, p = 0.044 < 0.05). However, we observed no significant change in the activated population activity for both *Cck*⁻/⁻ and Antagonist groups (*Figure 3F*; paired *t*-test, *Cck*⁻/⁻, Day 1 vs. Day 6, 0.0313 ± 0.0057 vs. 0.0386 ± 0.0099, p = 0.237; Antagonist, Day 1 vs. Day 6, 0.0218 ± 0.0094 vs. 0.0354 ± 0.0080, p = 0.240).

We adopted the Pearson correlation coefficient to evaluate the recurrence of neuronal activity among reaching trials. We compared the average correlation coefficient of neuronal activity of different trials between Days 1 and 6. We observed a significant increase in the trial-to-trial population activity correlation on Day 6 compared with Day 1 in the C57BL/6 mice group (*Figure 3G*, one-way RM ANOVA, Day 1 vs. Day 6, 0.023 ± 0.01 vs. 0.12 ± 0.04, $F[1,9] = 5.342$, p = 0.046 < 0.05; *Figure 3—figure supplement 2*). However, we observed no significant differences in the correlations between Days 1 and 6 in the *Cck*⁻/⁻ group, nor in the Antagonist group (*Figure 3G*; one-way RM ANOVA, *Cck*⁻/⁻, Day 1 vs. Day 6, 0.10 ± 0.07 vs. 0.07 ± 0.06, $F[1,6] = 0.073$, p = 0.796; Antagonist, Day 1 vs. Day 6, 0.12 ± 0.07 vs. 0.12 ± 0.05, $F[1,6] = 0.005$, p = 0.944; *Figure 3—figure supplement 2*). The pairwise Hausdorff distance of trajectories in C57BL/6 group decreased significantly with training, while no significant changes were observed in *Cck*⁻/⁻ or Antagonist group, suggesting that the population activity is in line with the changes of the variation of the trajectories during motor learning (*Figure 3H*; paired *t*-test, C57BL/6, Day 1 vs. Day 6, 0.6613 ± 0.017 vs. 0.5588 ± 0.0227 cm, p = 0.0075 < 0.01; *Cck*⁻/⁻, Day 1 vs. Day 6, 0.6787 ± 0.0470 vs. 0.6760 ± 0.0501 cm, p = 0.9219; Antagonist, Day 1 vs. Day 6, 0.7012 ± 0.0594 vs. 0.6712 ± 0.0659 cm, p = 0.5606). The trial-to-trial population activity correlation in Antagonist group on Day 1 appeared to be higher than that in C57BL/6 group. This might be due to the fact that the drug blocked the trial-to-trial learning on Day 1, suppressing the exploration of the optimal path and abandonment of bad movements that would otherwise occur in WT mice.

Taken together, CCK deficiency causes defects in neuronal refinement and the reproducibility of neuronal activity among different trials during motor skill learning.

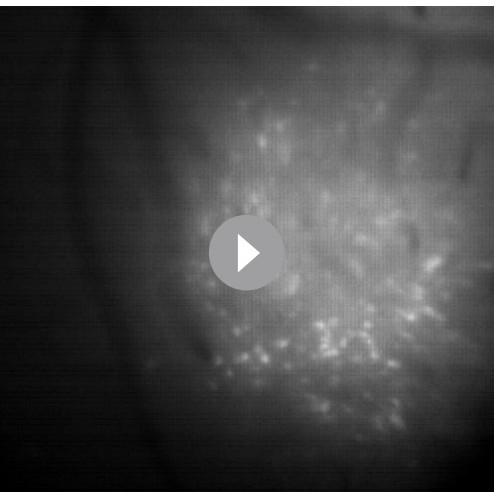

**Video 5.** Example of calcium signals of neurons in the motor cortex under the miniscope.
https://elifesciences.org/articles/83897/figures#video5

## CCK-expressing neurons in the rhinal cortex projecting to the motor cortex

Our next quest was to find what CCK projection is crucial in motor skill learning. We understand that CCK neurons in the entorhinal cortex, a gateway from the hippocampus to the neocortex, play critical roles in encoding sound–sound, visuo-auditory, fear, and spatial memory (*Li et al., 2014*; *Chen et al., 2019*; *Feng et al., 2021*; *Su et al., 2019*). These findings prompted us to examine whether CCK-expressing neurons in the entorhinal cortex also project to the motor cortex.

We used both anterograde and retrograde viruses to track neuronal projections in this study. We first injected a Cre-dependent, highly efficient AAV virus expressing mCherry into the rhinal cortex of one hemisphere of 8-week-old *Cck-Cre* mice (*Figure 4A*). This viral vector is expected to be taken up in the soma of neurons and transported to the axon terminus. In the motor cortex, mCherry-expressing neuronal axons mainly spread in layer 2/3 or 6 (*Figure 4B*). We next injected a Cre-dependent retrograde AAV vector expressing EYFP fluorescent protein gene into the motor cortex in deep layers and superficial layers to verify the projections from the rhinal cortex to the motor cortex (*Figure 4C*). In the rhinal cortex, the EYFP-labeled soma spread from AP: −2.54 mm to AP: −4.30 mm, and local clusters were observed in layers 3 and 5, where the neurons are expected to project to the neocortex (*Figure 4D*). Both anterograde and retrograde tracking results indicated that CCK-expressing neurons in the rhinal cortex projecting to the motor cortex were asymmetric, showing a preference for the ipsilateral hemisphere (*Figure 4—figure supplement 1*). Primary antibodies against GAD67 and CaMKIIa were used for immunostaining of the rhinal cortex sections to determine the characteristics of CCK neurons projecting to the motor cortex. None of the retrograde EYFP-labeled neurons merged with GAD67 staining but completely colocalized with CaMKIIa staining, indicated by the white arrowhead, suggesting that the neurons projecting to the motor cortex are all excitatory neurons (*Figure 4E, F*). Therefore, CCK neurons in the rhinal cortex may affect motor skill learning by regulating the plasticity of neurons in the motor cortex.

## Inhibiting CCK neurons in the RC suppresses motor learning

In the following experiment, we adopted chemogenetic method to selectively silence the CCK-positive neurons in the rhinal cortex to examine their involvement in motor skill learning.

We injected a Cre-dependent AAV vector carrying hM4Di or mCherry into the rhinal cortex bilaterally in *Cck-Cre* mice 1 month before the behavior test (*Figure 5A*). Clozapine was intraperitoneally injected, followed by an approximately 30 min period for drugs to be taken up and transported to the brain. The drug bound to the hM4Di and inactivated the neurons (*Figure 5A*). The success rate of hM4Di with the clozapine injection group showed no significant increase after 6 days of training, while the success rate of the control group of mCherry with clozapine injection increased significantly beginning on the third day of training and remained at a high level until the end of training (*Figure 5B*, *Figure 5—figure supplement 1A*, B; hM4Di + Clozapine group, one-way RM ANOVA, $F[5,50]$ = 0.839, p = 0.528; mCherry + Clozapine group, one-way RM ANOVA, $F[5,35]$ = 3.121, p = 0.02 < 0.05; two-way mixed ANOVA, post hoc comparison between two groups, $F[1,17]$ = 7.014, p = 0.016 < 0.05, hM4Di vs. mCherry, Day 3, 12.92% ± 3.10% vs. 25.99% ± 3.62%, $F[1,17]$ = 7.510, p = 0.014 < 0.05; Day 4, 12.04% ± 1.84% vs. 24.78% ± 3.34%, $F[1,17]$ = 12.804, p = 0.002 < 0.01; Day 5, 15.02% ± 2.55% vs. 25.74% ± 3.72%, $F[1,17]$ = 6.061, p = 0.025 < 0.05; Day 6, 14.41% ± 4.01% vs. 28.42% ± 5.64%, $F[1,17]$ = 4.354, p = 0.052.).

To exclude the possibility that hM4Di alone might regulate the neurons in this system, we administered saline rather than clozapine to *Cck-Cre* mice that had the same hM4Di viral vector injected into the rhinal cortex of *Cck-Cre* mice, as compared to the clozapine-administered experimental group (*Figure 5A*). The learning curve of the control group injected with saline showed a learning trend in the single pellet reaching task, similar to the 'mCherry + clozapine' group, and the success rate was significantly different from the 'hM4Di + clozapine' group (*Figure 5C*, *Figure 5—figure supplement 1C*; hM4Di + saline group, one-way RM ANOVA, $F[5,45]$ = 7.911, p < 0.001; between groups, two-way mixed ANOVA, significant interaction, $F[5,95]$ = 2.813, p = 0.021 < 0.05, hM4Di + saline vs. hM4Di + clozapne, post hoc comparison between two groups, $F[1,19]$ = 6.193, p = 0.022 < 0.05; post hoc comparison between two groups on different days, Day 3, 24.02% ± 3.93% vs. 12.12% ± 3.10%, $F[1,19]$ = 5.013, p = 0.0373 < 0.05; Day 4, 27.81% ± 3.84% vs. 12.04% ± 1.84%, $F[1,19]$ = 14.534, p = 0.0012 < 0.01; Day 5, 24.54% ± 3.05% vs. 15.02% ± 2.55%, $F[1,19]$ = 5.785, p = 0.0263 < 0.05; Day 6,

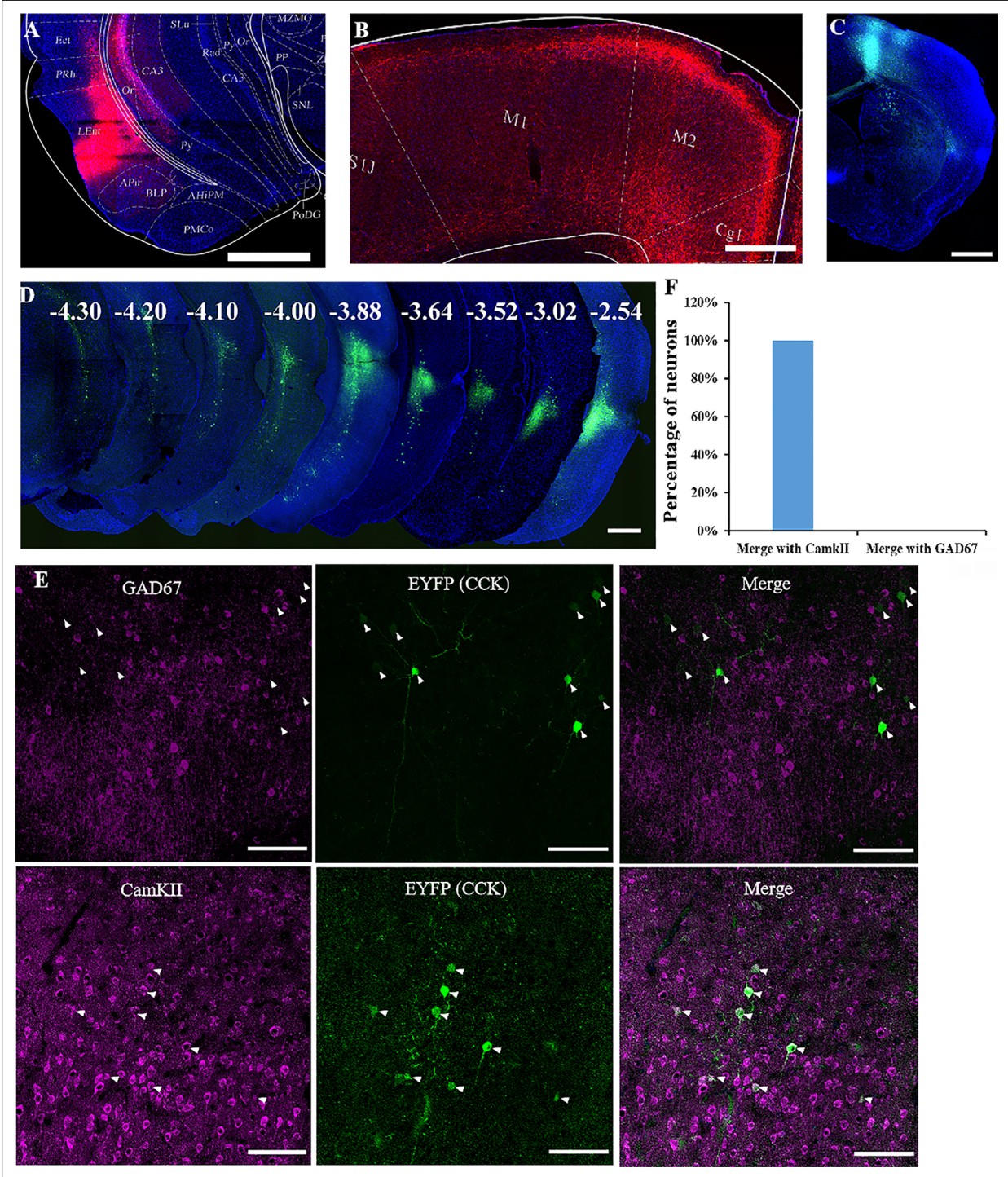

**Figure 4.** Labeling of cholecystokinin (CCK) neuron projections from the rhinal cortex (RC) to the MC. (**A**) Coronal section showing the virus injection site. The Cre-dependent AAV-hsyn-DIO-mCherry virus was injected into *Cck-Cre* mice. (**B**) Effective labeling of CCK neuron fibers in the MC. (**C**) Cre-dependent retrograde AAV virus injection site in the MC of the *Cck-Cre* mouse. (**D**) Continuous coronal brain sections showing EYFP in the lateral EC. The numbers (mm) indicate the position of the sections relative to the bregma. (**E**) GAD67 staining did not merge with the retrograde tracking CCK-positive neurons in the EC and CaMKII staining merged with the signal of retrograde tracking CCK neurons EC projecting. Arrowhead indicate the positions of CCK neurons. (**F**) Percentage of retrogradely labeled neurons merged with CamKII and GAD67 (*N* = 4, a total of 140 neurons for CamKII and 136 for GAD67). Scale bars represent 1000 μm in (**A**), (**B**), (**C**), and (**D**) and 100 μm in (**E**).

The online version of this article includes the following figure supplement(s) for figure 4:

**Figure supplement 1.** Three examples of CCK+ neurons in the RC of the *Cck-Cre* mouse injected with AAV(retro)-EF1a-DIO-EGFP in the MC (**A–C**).

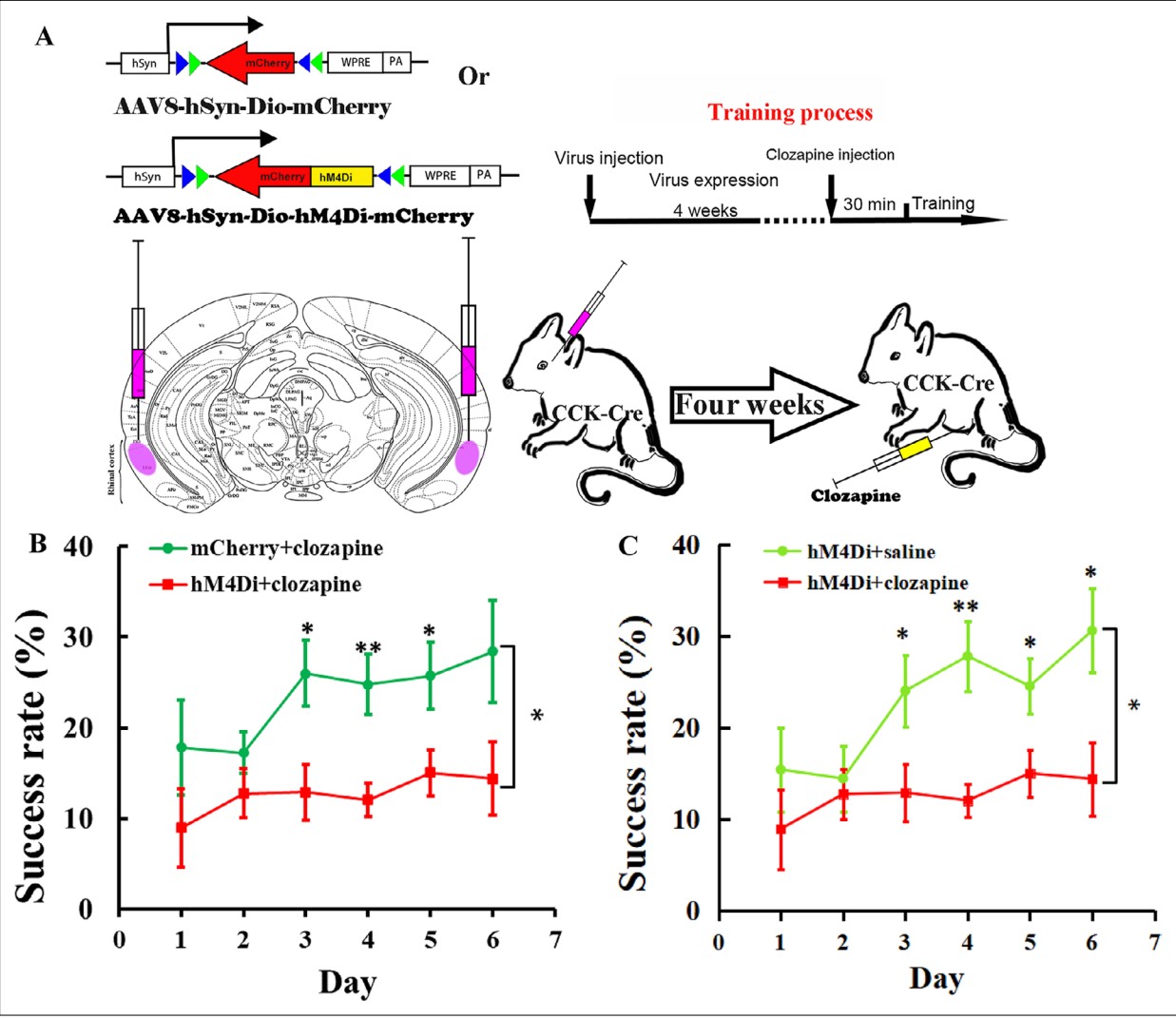

**Figure 5.** Effect of inhibition of the RC cholecystokinin (CCK) neurons on motor learning. (**A**) Experimental paradigm for the chemogenetic experiment. Cre-dependent AAV-DIO-hM4Di-mCherry or AAV-DIO-mCherry was infused into the rhinal cortex of *Cck-Cre* mice. After 4 weeks, clozapine or saline was intraperitoneally injected 30 min before training. (**B**) Success rate of *Cck-Cre* mice injected with hM4Di containing virus together with clozapine (hM4Di + clozapine) (*N* = 10) and control virus with clozapine (mCherry + clozapine) (*N* = 8). (**C**) Success rate of *Cck-Cre* mice injected with hM4Di containing virus plus clozapine (hM4di + clozapine, shared with (**B**)) and hM4Di plus saline (hM4Di + saline) (*N* = 11). The hM4Di + clozapine curve in **C** shared that in (**B**). *p < 0.05, **p < 0.01. Two-way mixed analysis of variance (ANOVA), post hoc comparison between two groups on different days.

The online version of this article includes the following figure supplement(s) for figure 5:

**Figure supplement 1.** Learning curve of single *Cck-Cre* mouse injected with hM4Di-clozapine (**A**), Control-clozapine (**B**), and hM4Di-saline (**C**), and example of the expression of hM4Di virus in the rhinal cortex (**D**).

30.60% ± 4.59% vs. 14.41 ± 4.01%, *F*[1,19] = 7.128, p = 0.0151 < 0.05; The hM4Di + clozapine curve in *Figure 5C* shared that in *Figure 5B*). These results concluded that CCK neurons in the rhinal cortex may be crucial for motor learning.

## Specifically inhibiting the projection from RC to the MC suppressed the motor learning

To test the percentage of CCK+ and CCK− neurons in the RC that project to the MC, we injected the AAV(retro)-EF1a-DO-mCherry-DIO-EGFP virus in the MC of *Cck-Cre* mice and observed well-expressed virus in the MC and RC (*Figure 6A*, *Figure 6—figure supplement 1A, B*). We found 98.67 ± 1.33% neurons are EGFP-positive and 1.33% ± 1.33% neurons are mCherry-positive, indicating that almost all neurons from the RC to the MC are CCK-positive (*Figure 6A, B*). Besides, we also

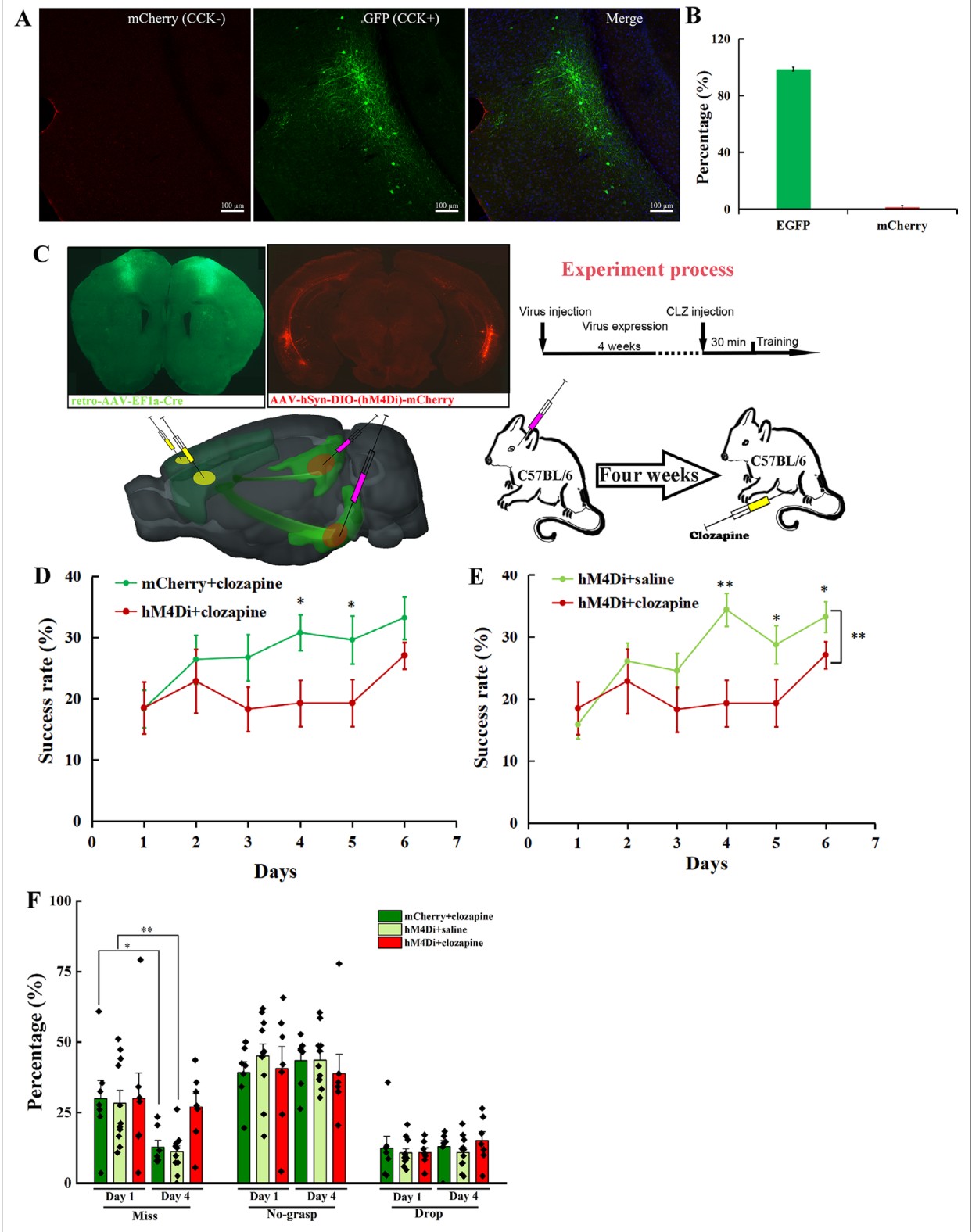

**Figure 6.** Effect of inhibition of neurons projecting from the RC to the MC on motor learning. (**A**) Retrograde neurons in the EC of *Cck-Cre* mice injected with AAV(retro)-EF1a-DO-mCherry-DIO-EGFP in the MC. (**B**) Percentage of EGFP (CCK+) and mCherry (CCK−) in the EC. *N* = 3, *n* = 10, a total of 111 neurons. (**C**) Experimental paradigm for the chemogenetic experiment. A retro-Cre virus was injected in the MC of two hemispheres and a Cre-dependent hM4Di (or not, as a negetive control) virus was injected in the EC. After 4 weeks of expression, mice were trained to learn the single pellet reaching task. Thirty minutes before training, mice were injected with clozapine (i.p.) every day. (**D**) Success rate of mice injected with hM4Di containing

*Figure 6 continued on next page*

*Figure 6 continued*

virus together with clozapine (hM4Di + clozapine) (N = 7) and control virus with clozapine (mCherry + clozapine) (N = 8). (**E**) Success rate of mice injected with hM4Di containing virus plus clozapine (hM4di + clozapine, shared with (**D**)) and hM4Di combined with saline (hM4Di + saline) (N = 11). The hM4Di + clozapine curve in *Figure 5C* shared that in (**D**). (**F**) Detailed reaching results for the three different treatments on Days 1 and 4. *p < 0.05, **p < 0.01. Two-way mixed analysis of variance (ANOVA), post hoc comparison between two groups on different days.

The online version of this article includes the following figure supplement(s) for figure 6:

**Figure supplement 1.** Experimental paradigm of AAV(retro)-EF1a-DO-mCherry-DIO-EGFP injected in the MC of the *Cck-Cre* mice (**A**).

**Figure supplement 2.** AAV(retro)-hSyn-mCherry injected in the MC of *Cck-Cre* mice.

**Figure supplement 3.** Learning curve of single C57BL/6 mouse injected with Cre-dependent ± hM4Di (EC) and retro-Cre (MC, two sides).

**Figure supplement 4.** Expression of retro-Cre in the single side of the MC (**A**) and Cre-dependent hM4Di-mCherry in th RC (**B**).

**Figure supplement 5.** Learning curve of mice injected with hM4Di (SMC, retro-Cre in the single motor cortex contralateral to the dominant hand and hM4Di in the entorhinal cortex) and clozapine compared with mice injected with control virus and clozapine (**A**, shared with *Figure 6B*), hM4Di and saline (**B**, shared with *Figure 6C*), and learning curve of single mouse (**C**).

immunostained the Cre in the RC of *Cck-Cre* mice injected with a retro-mCherry virus in the MC to verify the above conclusion. Over 96% of the retrograde neurons are Cre-positive, indicating that neurons in the RC projecting to the MC are CCK-positive (*Figure 6—figure supplement 2A, B*).

Therefore, we next bilaterally injected a retro-Cre virus into the MC and a Cre-dependent hM4Di virus in the RC of C57BL/6 mice to specifically inhibit the CCK-positive neurons projecting to the MC (*Figure 6C*). After 4 weeks, the retro-Cre (EGFP) and hM4Di (mCherry) viruses were well expressed in the MC and RC, respectively (*Figure 6C*). Clozapine was injected (i.p.) 30 min before training every day. Additionally, two negative control groups were set. One was the Cre-dependent hM4Di virus combined with saline (hM4Di + saline) to rule out the effect of hM4Di, the other was control virus together with clozapine (mCherry + clozapine) to exclude the effect of clozapine. Compared with the two control groups, the success rate of the hM4Di + clozapine group remained at the original level until Day 5, and started to increase on Day 6, while for two control groups, the success rate gradually increased and was significantly higher than that of the 'hM4Di + clozapine' group, indicating that specifically inhibiting the projections from the RC to the MC significantly suppressed motor learning ability (*Figure 6D, E*, *Figure 6—figure supplement 3A–C*; mCherry + clozapine vs. hM4Di + clozapine, two-way mixed ANOVA, $F[1,13] = 3.893$, p = 0.070; post hoc comparison between two groups, mCherry + clozapine vs. hM4Di + clozapine, Day 3, 26.76% ± 3.79% vs. 18.31% ± 3.62%, p = 0.07, Day 4, 30.81% ± 2.95% vs. 19.31% ± 3.79%, p = 0.018, Day 5, 29.64% ± 3.97% vs. 19.32% ± 3.79%, p = 0.044, Day 6, 33.24% ± 3.49% vs. 27.09% ± 2.17%, p = 0.091; pairwise comparison between different days, mCherry + clozapine, Day 1 vs. Day 3, 18.37% ± 3.10% vs. 26.76% ± 3.79%, p = 0.171, Day 1 vs. Day 4, 18.37% ± 3.10% vs. 30.81% ± 2.95%, p = 0.006, Day 1 vs. Day 5, 18.37% ± 3.10% vs. 29.64% ± 3.97%, p = 0.006, Day 1 vs. Day 6, 18.37% ± 3.10% vs. 33.24% ± 3.49%, p = 0.0001, hM4Di + clozapine, Day 1 vs. Day 3, 18.52% ± 4.27% vs. 18.31% ± 3.62%, p = 0.612, Day 1 vs. Day 4, 18.52% ± 4.27% vs. 19.31% ± 3.79%, p = 0.826, Day 1 vs. Day 5, 18.52% ± 4.27% vs. 19.32% ± 3.79%, p = 0.800, Day 1 vs. Day 6, 18.52% ± 4.27% vs. 27.09% ± 2.17%, p = 0.128; hM4Di + saline vs. hM4Di + clozapine, two-way mixed ANOVA, $F[1,17] = 5.239$, p = 0.035, post hoc comparison between two groups, hM4Di + saline vs. hM4Di + clozapine, Day 3, 24.54% ± 2.89% vs. 18.31% ± 3.62%, p = 0.0895, Day 4, 34.40% ± 2.69% vs. 19.31% ± 3.79%, p = 0.0016, Day 5, 28.74% ± 3.06% vs. 19.32% ± 3.79%, p = 0.0296, Day 6, 33.24% ± 2.46% vs. 27.09 ± 2.17%, p = 0.0424; pairwise comparison between different days, hM4Di + saline, Day 1 vs. Day 3, 15.87% ± 2.29% vs. 24.54% ± 2.89%, p = 0.0192, Day 1 vs. Day 4, 15.87% ± 2.29% vs. 34.40% ± 2.69%, p = 0.0009, Day 1 vs. Day 5, 15.87% ± 2.29% vs. 28.74% ± 3.06%, p = 0.0024, Day 1 vs. Day 6, 15.87% ± 2.29% vs. 33.24% ± 2.46%, p < 0.001). The gradual increase on Day 6 may be due to the incomplete inhibition of the projection from the RC to the MC (*Figure 6D, E*). We also compared the detailed reaching results of the three groups at Days 1 and 4. In terms of miss rate, the 'hM4Di + clozapine' group had no significant change, while two control groups dropped significantly, suggesting that the aiming and advance learning ability were suppressed by the inhibition of the projections from the RC to the MC (*Figure 6F*; paired *t*-test, miss, Day 1 vs. Day 4, hM4Di + clozapine, 30.01% ± 9.08% vs. 27.06% ± 4.67%, p = 0.78, mCherry + clozapine, 30.00% ± 6.45% vs. 12.74% ± 2.48%, p = 0.91, hM4Di + saline, 28.40% ± 4.48% vs. 11.11% ± 2.12%, p = 0.95). Like other treatments, the no-grasp rate and drop rate of the 'hM4Di +

clozapine' group had no significant change (*Figure 6F*; paired *t*-test, no-grasp, Day 1 vs. Day 4, hM4Di + clozapine, 40.61% ± 7.88% vs. 38.83% ± 6.84%, p = 0.87, mCherry + clozapine, 39.21% ± 3.84% vs. 43.47% ± 3.50%, p = 0.43, hM4Di + saline, 45.02% ± 4.29% vs. 43.64% ± 2.99%, p = 0.79; drop, Day 1 vs. Day 4, hM4Di + clozapine, 10.86% ± 1.70% vs. 15.15% ± 3.10%, p = 0.25, mCherry + clozapine, 12.42% ± 4.19% vs. 12.98% ± 2.27%, p = 0.91, hM4Di + saline, 10.71% ± 1.48% vs. 10.86% ± 1.73%, p = 0.95). To exclude the possibility that inhibiting the projection from the RC to the MC impaired the basic movement ability, we injected (i.p.) clozapine to mice with hM4Di after they learned the single pellet reaching task. In terms of 'miss', 'no-grasp', 'drop', and 'success', there were no significant differences between before and after clozapine injection (*Figure 6—figure supplement 3D*; paired *t*-test, before vs. after, miss, 7.43% ± 2.48% vs. 6.74% ± 2.20%, no-grasp, 44.07% ± 2.86% vs. 41.10% ± 1.61%, drop, 10.41% ± 2.95% vs. 13.53% ± 1.79%, success, 38.10% ± 2.20% vs. 38.64% ± 2.27%).

To test whether the MC in both hemispheres is involved in motor skill learning, we also injected a retro-Cre virus in the MC contralateral to the dominant forelimb (hM4Di(SMC) + clozapine). Viruses were well expressed in the MC and the RC (*Figure 6—figure supplement 4A, B*). The motor learning ability was suppressed slightly by inhibiting the activity of the projection from the RC to the MC contralateral to the dominant forelimb as the success rate of the 'hM4Di(SMC) + clozapine' group increased to the plateau on Day 5, though delayed by 1 day compared to the two control groups (*Figure 6—figure supplement 5A–C*; paired *t*-test, Day 1 vs. Day 3, 19.73% ± 1.56% vs. 26.09% ± 3.80%, p = 0.1368, Day 1 vs. Day 4, 19.73% ± 1.56% vs. 25.29% ± 2.38%, p = 0.0862, Day 1 vs. Day 5, 19.73% ± 1.56% vs. 29.50% ± 2.52%, p = 0.0295, Day 1 vs. Day 6, 19.73% ± 1.56% vs. 30.40% ± 3.49%, p = 0.0476). Besides, compared with two control groups (shared with *Figure 6*), the learning ability was not significantly suppressed, indicating that the projections from the RC to the bilateral MC play a critical role in motor skill learning and the ipsilateral MC may compensate the contralateral MC (*Figure 6—figure supplement 5A, B*; mCherry + clozapine vs. hM4Di(SMC) + clozapine, two-way mixed ANOVA, $F[1,15]$ = 0.313, p = 0.584, post hoc comparison two groups, Day 3, 26.76% ± 3.79% vs. 26.09% ± 3.80%, p = 0.90, Day 4, 30.81% ± 2.95% vs. 25.29% ± 2.38%, p = 0.16, Day 5, 29.64% ± 3.97% vs. 29.50% ± 2.52%, p = 0.98, Day 6, 33.24% ± 3.49% vs. 30.40% ± 3.86%, p = 0.61; hM4Di + saline vs. hM4Di(SMC) + clozapine, two-way mixed ANOVA, $F[1,19]$ = 0.278, p = 0.604, post hoc comparison between two groups, Day 3, 24.54% ± 3.88% vs. 26.09% ± 3.80%, p = 0.747, Day 4, 34.40% ± 4.08% vs. 25.29% ± 2.38%, p = 0.021, Day 5, 28.74% ± 4.07% vs. 29.50% ± 2.52%, p = 0.852, Day 6, 33.24% ± 3.58% vs. 30.40% ± 3.86%, p = 0.535).

Taken together, these results suggested that bilateral projections of CCK neurons from the RC to the MC are crucial for motor skill learning.

## Rescue of the motor learning ability of the Cck⁻/⁻ mice with CCK4

So far, we have examined the potential involvement of CCK in motor skill learning with several loss-of-function studies. We next designed a gain-of-function experiment to see whether CCKBR agonist could rescue the defective motor learning ability. A tetrapeptide, CCK4 (Trp-Met-Asp-Phe-NH2), a CCKBR agonist that can pass through the brain–blood barrier, was chosen to regain the defective motor learning ability of *Cck*⁻/⁻ mice (*Feng et al., 2021*).

Firstly, we examined whether CCK4 could rescue the defective neuroplasticity in the motor cortex of *Cck*⁻/⁻ mice. We carried out electrophysiology recording on the motor cortex of the brain slices from the *Cck*⁻/⁻ mice. After 15 min of stable baseline recording, CCK4 or vehicle was injected into the electrode dish and applied HFS, followed by 60 min of recording. We observed a significant rescuing effect by CCK4 application before the HFS compared with its vehicle control (*Figure 7A, B*; Vehicle vs. CCK4, two-way mixed ANOVA, significant interaction during −10 to 0 min and 50 to 60 min, $F[1,21]$ = 10.656, p = 0.004 < 0.01; post hoc comparison between two groups, $F[1,21]$ = 7.997, p = 0.01 < 0.05; Vehicle, before vs. after, 100.95% ± 0.67% vs. 95.53% ± 5.77%, $F[1,10]$ = 1.239, p = 0.292; CCK4, before vs. after, 100.28% ± 0.47% vs. 118.89% ± 6.09%, $F[1,11]$ = 11.653, p = 0.006 < 0.01).

Next, we examined whether the CCK4 application could rescue the motor skill learning of *Cck*⁻/⁻ mice. We injected with CCK4 or vehicle solution intraperitoneally to *Cck*⁻/⁻ mice every day before the 6-day training (*Figure 7C*). The success rate of CCK4-injected group kept at the baseline level in the first 3 days and started to increase gradually from Days 4 to 6 (*Figure 7D*, *Figure 7—figure supplement 1A*; CCK4, one-way RM ANOVA, $F[5,50]$ = 3.914, p = 0.005 < 0.01; Day 5 vs. Day 1, 30.58% ± 4.18% vs. 19.17% ± 3.03%, $F[1,10]$ = 5.680, p = 0.038 < 0.05; Day 6 vs. Day 1, 31.50% ± 4.43% vs.

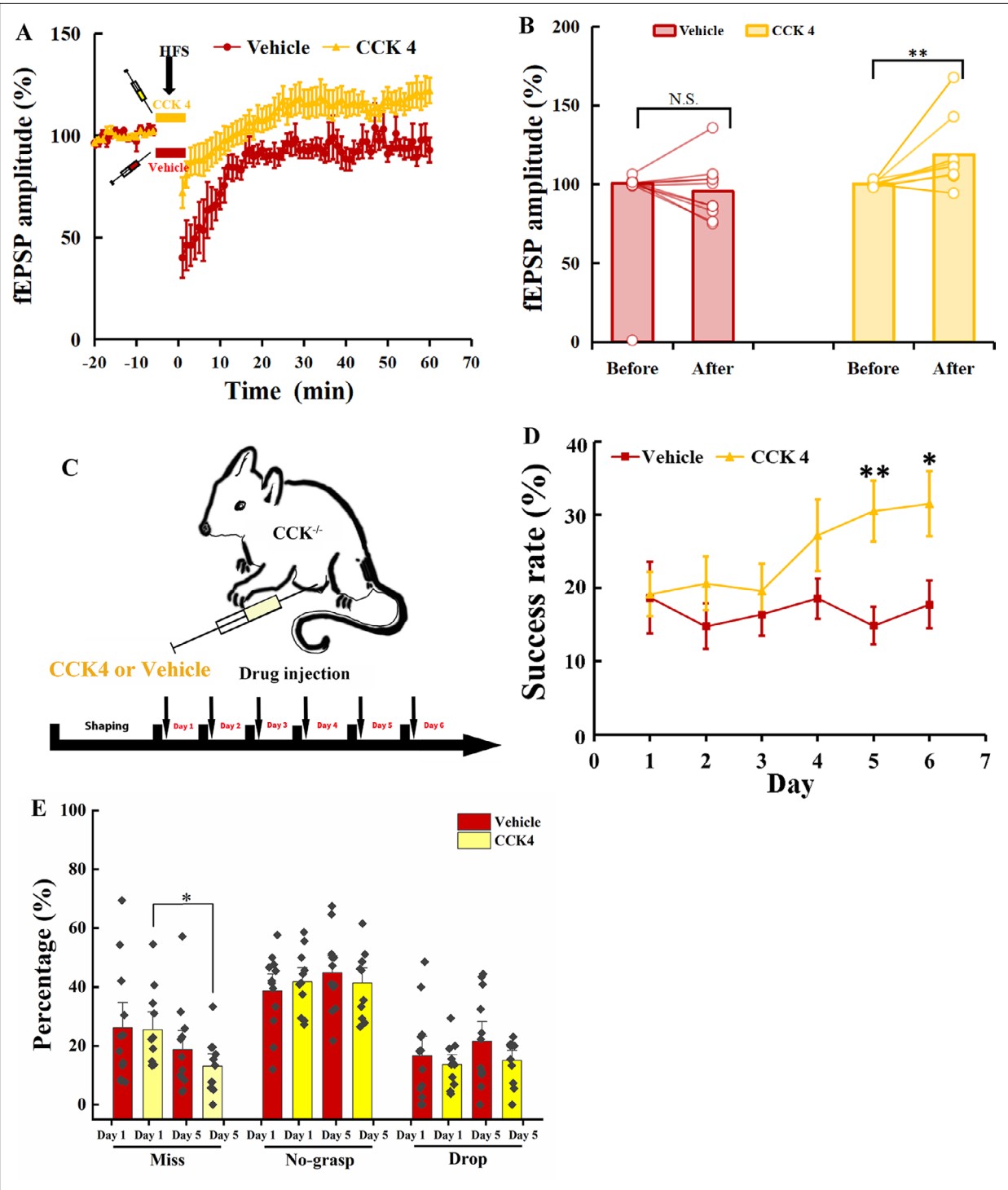

**Figure 7.** Rescuing the motor learning ability of *Cck*[−/−] mice by CCK4. (**A**) Normalized field excitatory postsynaptic potential (fEPSP) amplitude before and after high-frequency stimulation (HFS) of the MC of *Cck*[−/−] mice applied with CCK4 (*N* = 6, *n* = 14) or vehicle (*N* = 6, *n* = 11). (**B**) The average normalized fEPSP amplitude 10 min before HFS (−10 to 0 min, before) and 10 min after HFS (50 to 60 min, after) in the MC of *Cck*[−/−] mice injected with CCK4 or vehicle. *p < 0.05, **p < 0.01. Two-way mixed analysis of variance (ANOVA) with Bonferroni pairwise comparison. (**C**) Experimental paradigm for cholecystokinin (CCK) rescuing experiment. CCK4 or vehicle was injected (i.p.) every day before training. (**D**) Success rate of *Cck*[−/−] mice injected with CCK4 (*N* = 11) or vehicle (*N* = 10). *p < 0.05, **p < 0.01. Two-way mixed ANOVA, post hoc comparison between two groups on Days 5 and 6. (**E**) Detailed reaching results or *Cck*[−/−] mice injected (i.p.) with vehicle and CCK4 on Days 1 and 5. *p < 005, N.S., not significant. Paired *t*-test.

The online version of this article includes the following figure supplement(s) for figure 7:

*Figure 7 continued on next page*

*Figure 7 continued*

**Figure supplement 1.** Learning curve of single $Cck^{-/-}$ mouse administrated with Vehicle (**A**) and CCK4 (**B**), and comparison between $Cck^{-/-}$ mice injected with CCK4 and WT group (**C**) and learning curve of $Cck^{-/-}$ mice injected with CCK8 in the motor cortex, for the average (**D**) and every single mouse (**E**).

19.17% ± 3.03%, $F[1,10] = 6.893$, p = 0.025 < 0.05). In contrast, we observed no improvement in the success rate in the Vehicle group (*Figure 7D*, *Figure 7—figure supplement 1B*; Vehicle, one-way RM ANOVA, $F[5,55] = 0.476$, p = 0.793). The between-group comparison showed that the CCK4 group had significantly higher success rate from Days 5 to 6 compared to the Vehicle group (*Figure 7D*; Vehicle vs. CCK4, two-way mixed ANOVA, significant interaction, $F[5,105] = 2.405$, p = 0.043 < 0.05; post hoc comparison between Vehicle and CCK4, Day 5, 14.88% ± 2.61% vs. 30.51% ± 4.18%, $F[1,21] = 10.459$, p = 0.004 < 0.01; Day 6, 17.76% ± 3.25% vs. 31.50% ± 4.43%; $F[1,21] = 6.412$, p = 0.019 < 0.05).

We compared the detailed reaching results on Days 1 and 5 between the CCK4 and the Vehicle groups. We found the miss rate of the CCK4 group dropped significantly at Day 5 compared to Day 1, while that of the Vehicle group showed no significant change (*Figure 7E*; paired *t*-test, Vehicle, Day 1 vs. Day 5, 26.12% ± 5.71% vs. 18.71% ± 4.31%; $F[1,11] = 1.155$, p = 0.305; CCK4, Day 1 vs. Day 5, 25.47% ± 4.03% vs. 13.13% ± 2.80%, $F[1,10] = 6.643$, p = 0.028 < 0.05), suggesting that the CCK4 rescued the aiming in reaching. This result demonstrated that CCK4 could cross the brain–blood barrier and partially rescue the motor learning ability of $Cck^{-/-}$ mice (*Figure 7—figure supplement 1C*, two-way mixed ANOVA, CCK4 vs. WT, $F[1,19] = 0.267$, p = 0.612; *t*-test, CCK4 vs. WT, Day 3, 19.63% ± 3.68% vs. 30.94% ± 4.17%, p = 0.055, Day 4, 27.20% ± 4.86% vs. 28.96% ± 2.90%, p = 0.765, Day 5, 30.51% ± 4.18% vs. 31.90% ± 3.50%, p = 0.805, Day 6, 31.50% ± 4.43% vs. 32.76% ± 3.12%, p = 0.822).

Therefore, CCK is the crucial signal that enables motor learning. Intraperitoneal injection of CCK4 is sufficient to rescue the motor learning ability by turning on the neuroplasticity of the $Cck^{-/-}$ mice.

## Discussion

$Cck^{-/-}$ mice showed defective motor learning ability, of which the success rate of retrieving reward remained at the baseline level compared to the WT mice with a significantly increased success rate. We induced no LTP by HFS in the motor cortex of $Cck^{-/-}$ mice but readily in their WT control, indicating a possible association between the motor learning deficiency and neuroplasticity in the motor cortex. In vivo calcium imaging demonstrated that the deficiency of CCK signaling led to the defect in the population neuronal plasticity in the motor cortex affecting motor skill learning.

We found that the CCK-positive neurons in the rhinal cortex projected to the motor cortex, using both anterograde and retrograde tracing methods. Inactivating the CCK neurons in the rhinal cortex or specifically inactivating the CCK projections from the rhinal cortex to the motor cortex using chemogenetic methods significantly suppressed the motor learning ability. Our further gain-of-function study revealed that intraperitoneal application of CCK4 rescued the defective motor skill learning of $Cck^{-/-}$ mice.

### CCK is involved in motor skill learning

Neuroplasticity of the motor cortex has been assessed by many researchers using multiple methods, such as single pellet reaching task and lever-press task (*Xu et al., 2009*; *Peters et al., 2014*). Other brain areas are also involved in motor skills learning, such as thalamus, striatum, cerebellum, and midbrain. Thalamocortical projections in the motor cortex are widely distributed in all layers, including inputs to corticospinal neurons in layer 5 (*Hooks et al., 2013*). With single pellet reaching task training, thalamocortical neurons are biased in activating the corticospinal neurons that control the performance of the task, though the unbiased activation of corticospinal neurons was observed before training, suggesting that the thalamus selectively activates corticospinal neurons to generate better control of the forelimb movement with motor learning (*Biane et al., 2016*). The spiking of Purkinje neurons switched from more autonomous, the baseline condition, to time-locked activation or silence before reaching onset to produce a state promoting a high quality of movement, as mice learn to direct a robotic manipulation toward a target zone (*Wagner et al., 2021*). The ventral tegmental

area (VTA) dopaminergic projection in the motor cortex is necessary for motor skill learning but not for execution. The VTA projection to the motor cortex may facilitate the encoding of a motor skill memory by relaying food reward information related to the task (*Hosp et al., 2011*). As the core area where dexterous motor memory is encoded, the plasticity of the motor cortex enables animals to learn complex motor tasks.

CCK produced in the rhinal cortex has been identified as the key to transforming a paired tone into auditory memory in mice and rats by regulating the plasticity of the auditory cortex (*Li et al., 2014*). In the present study, we found that genetic knockout of the *Cck* gene caused defects in motor learning, while the success rate of WT mice increased to 30.94% on Day 3. The success rate alone is not sufficient to describe the function of CCK in motor skill learning; therefore, the reaching result of the task is divided into four types, 'miss', 'no-grasp', 'drop', and 'success'. 'Miss' is caused by defects in 'aiming' and 'advance', indicating a low probability of hitting the pellet. Miss rate of the $Cck^{-/-}$ mice had no significant change with learning and showed significant difference for the WT mice, suggesting that the brain areas controlling the 'aiming' and 'advance' are affected by CCK defect. The lack of CCK impaired the plasticity of the motor cortex, which is deemed the basis for motor learning.

*Duchemin et al., 1987* studied the *Cck* gene expression level in the brain of rats pre- and post-natally. They found that the CCK mRNA was detectable on embryonic day 14 (E14) and gradually increased to the maximum level on postnatal day 14 (P14). *Giacobini and Wray, 2008* mapped the expression of CCK in the mouse brain. Plentiful CCK expression was observed at E12.5 in the thalamus and spinal cord, and by E17.5, CCK expression extended to the cortex, hippocampus, and hypothalamus, suggesting that CCK might participate in the development of rats and mice. Besides, evidence also showed that CCK suppressed the migration of gonadotropin-releasing hormone-1 (GnRH-1) through CCK-A receptor in the brain of mice, and CCK assisted lambs get mother preference at the early time, indicating that CCK might play a role in development (*Nowak et al., 2001*; *Giacobini and Wray, 2007*). In this study, we found the basic movement ability of $Cck^{-/-}$ and WT mice, including stride length, stride time, step cycle ratio, and grasp force, was comparable (*Figure 1—figure supplement 1C–F*), suggesting that knockout of *Cck* gene did not affect the basic movement ability. This could be because the development of basic movement ability is not learning guided, but is physical structure determined.

The motor cortex plays the leading role in controlling motor memory encoding (*Cheney, 1985*; *Sanes and Donoghue, 2000*; *Economo et al., 2018*; *Svoboda and Li, 2018*). CCKBRs dominate CCKARs in the neocortex including the motor cortex (*Crawley and Corwin, 1994*; *Wank, 1995*). Blockade of the CCKBRs in the motor cortex suppressed the improvement in the success rate of mice in the single pellet reaching task (*Figure 2B*, *Figure 2—figure supplement 1D, E*). The motor learning ability of Antagonist group was comparable to $Cck^{-/-}$ mice, suggesting that the role CCK played in the motor cortex is the key to motor skill learning. The gradually and slightly increased success rate on Days 5 and 6 (*Figure 2B*) could be attributed to two possible reasons: the antagonists might not last long enough to cover the whole training period every day; the formation of glia scar around the cannula could block the spreading of the drug. Besides, CCKBR antagonist did not affect the learned skill of the task, suggesting that CCKBR antagonist suppressed motor skill learning not by impairing the movement ability of mice (*Figure 2—figure supplement 1D*).

## CCK neural projections from the rhinal cortex to the motor cortex regulating neruoplasticity in the motor cortex

Based on the evidence that CCK is important for neuroplasticity of the motor cortex and motor skill learning, the next question is how CCK affects the changes in neural activity of the motor cortex during training. An earlier study found that neural activity in layer 2/3 of the motor cortex was modified, exhibiting more reproducible spatiotemporal sequences of neuronal activity with motor learning (*Peters et al., 2014*).

In the present study, the neural activity related to the task in layer 2/3 of the motor cortex of C57BL/6 mice was refined with motor skill learning, the activation of neurons becoming more reproducible among trials. The reproducibility changes of neural activity are in parallel with the reduced variations in the trajectories of the C57BL/6 mice after training (*Figure 1F, G*). However, $Cck^{-/-}$ mice generated distinct changes in the neuronal activity in the motor cortex compared with C57BL/6 mice. The pattern of the peak activity and the trial-to-trial population correlation had no significant

differences after 6 days of motor learning, suggesting no refinement in the neuronal circuit after motor learning in *Cck*$^{-/-}$ mice (*Figure 3D*).

In order to exclude a different background of neuronal activity due to long-term accommodation to the lack of CCK in *Cck*$^{-/-}$ mice, we injected the CCKBR antagonist, L365.260, into the motor cortex of C57BL/6 mice and observed no significant changes in the pattern of the peak activity and the trial-to-trial population correlation had after 6 days of motor learning, similar to the *Cck*$^{-/-}$ mice.

The entorhinal cortex is crucial for learning and memory (*Chen et al., 2013*; *Feng et al., 2021*). Our group found that CCK is essential for neuroplasticity in the auditory cortex (*Li et al., 2014*). In this research, we determined that CCK from the rhinal cortex may be critical for motor skill learning.

In the rhinal cortex, CCK-positive neurons that project to the motor cortex are excitatory neurons (*Figure 4E, F*). The roles of both CCK and glutamate in the neuroplasticity and the relationship between CCK and glutamate have been studied before (*Bandopadhyay and De Belleroche, 1991*; *Chen et al., 2019*). In the previous study, we found that CCK is critical for HFS-induced LTP, and CCK release is triggered by the activation of NMDA receptors that could be located in the presynaptic membrane of CCK-positive neurons (*Chen et al., 2019*).

In the motor cortex, many CCK-positive neurons are γ-aminobutyric acid-ergic (GABAergic) neurons, in which the role played by CCK is not very clear (*Whissell et al., 2015*). However, evidence showed that GABA may inhibit the release of CCK in the neocortex (*Yaksh et al., 1987*). In this study, we found that a CCK antagonist injected locally into the motor cortex disrupted neuroplasticity and motor learning. While this effect is similar to that of chemogenetic inhibition of RC-to-MC CCK+ neurons, we cannot rule out the possibility that this CCK antagonist might also affect CCK+ inhibitory neurons locally in the motor cortex. In addition, many glutamatergic neurons in the neocortex also express CCK (*Watakabe et al., 2012*). Future studies are needed to investigate the role of motor cortical CCK-positive neurons, including inhibitory and excitatory neurons, played in neuroplasticity and motor skill learning.

The hippocampus system, including the rhinal cortex, plays an essential role in declarative learning based on the finding of the famous patient H.M. (*Corkin, 1968*). However, the understanding of the role of the hippocampus system in motor skills learning is not consistent (*Corkin, 1968*; *De Brigard, 2019*). In the mirror tracking task, the performance of H.M. was on par with normal people, suggesting that the motor learning ability was not affected without the hippocampus system (*Corkin, 1968*). But in the other two motor learning tasks, rotary pursuit and bimanual tracking, the performance of H.M. was much worse than the control. Besides, the movement of H.M. was slower when performing the task. Indeed, Corkin herself thought that the H.M. could perform tasks that required less demanding motor skills, but not the tasks demanding better motor skills (*Corkin, 1968*; *De Brigard, 2019*).

The single pellet reaching task is a complex and dexterous motor task requiring the motor cortex, hippocampus, thalamus, and the whole motor system. Chemogenetic inactivation of CCK neurons in the rhinal cortex significantly impaired the mice's motor learning ability compared to the two control groups. Besides, the specificity of the projections from the rhinal cortex to the motor cortex for motor skill learning was studied. Over 98% of neurons in the rhinal cortex that projected to the motor cortex are CCK positive (*Figure 6A*, *Figure 6—figure supplement 1A, B*), enabling us to inhibit CCK projections from the rhinal cortex to the motor cortex specifically by injecting the retro-Cre virus in the motor cortex and the Cre-dependent hM4Di in the rhinal cortex in C57BL/6 mice. Compared to two control groups, the learning ability of the experimental group was significant suppressed, suggesting that CCK projections from the rhinal cortex to the motor cortex are critical for motor skill learning (*Figure 6*). Whereas, specifically inhibiting the projections from the rhinal cortex to the motor cortex contralateral to the dominant forelimb alone is not enough to significantly suppress the motor learning ability, suggesting that the bilateral motor cortex is involved in motor skill learning and the ipsilateral motor cortex may compensate the contralateral motor cortex. This is consistent with the previous finding that the number of increased synaptic GluA1 in the ipsilateral motor cortex after motor skill learning is comparable to that in the contralateral motor cortex (*Roth et al., 2020*). Whereas, the result was contradictory to the result of antagonist manipulation, which could be because when the antagonist was locally infused in the contralateral motor cortex through a drug cannula, it might have diffused to the ipsilateral motor cortex. Another possibility is that locally infused CCK antagonist might have inhibited CCK released from other cell types, in addition to RC-to-MC projections, resulting in a more severe phenotype difficult to compensate.

The motor cortex is considered as the main area generating motor memory when training in the single pellet reaching task (*Komiyama et al., 2010 Peters et al., 2014*; *Dhawale et al., 2021*). Our results suggest that the plasticity of the motor cortex is important for motor learning, but did not imply that other brain areas which also receive CCK-positive neural projections from the rhinal cortex are not important for the performance of this task, such as the hippocampus, visual cortex, and basal ganglia. The hippocampus plays a critical role in reactivation of motor memories in the quiet-rest period in motor sequence learning and the dysfunctional dentate gyrus caused reduced success rate everyday but the learning rate is comparable to the control group in the single pellet reaching task (*Jacobacci et al., 2020*; *Hong et al., 2007*). The visual cortex was critical for mice to perform the single pellet reaching task when light was available, whereas visual cortex was not required when trained in the dark condition (*Roth et al., 2020*). The basal ganglia, considered to control action selection or vigor modulation, specifies and controls the learned skills (*Dhawale et al., 2021*). However, the role played by CCK in these brain areas during motor learning is unknown. The motor cortex was selected in this study because the motor cortex is the hub connected to various movement-related brain areas, such as the visual cortex and the basal ganglia, and integrates different information for the generation and execution of motor memories.

Based on the anterograde and retrograde tracing of the neurons in the rhinal cortex, terminals of projections from the rhinal cortex to the motor cortex were distributed to the superficial and deep layers (*Figure 4B, D*). Previous research on both layers 2/3 and 5 found that motor skill learning refined neuronal activity in layer 2/3 of the motor cortices of the mice in a lever-press task. Thus, the CCK projections in the superficial layer may be where plasticity occurs (*Peters et al., 2017*; *Heindorf et al., 2018*). Two-photon calcium imaging results from previous research indicated that spine generation and elimination occurred in the apical dendrites (in the superficial layer) of neurons in layer 2/3 (*Chen et al., 2015*). Still, the spines around the soma of the neurons in layer 2/3 showed no significant changes (*Chen et al., 2015*), consistent with the location of CCK neuron terminals projecting from the rhinal cortex.

## Rescuing neuroplasticity and motor skill learning using CCK4

Our gain-of-function experiment by injecting CCK4 to rescue the defective learning ability of $Cck^{-/-}$ mice supported the critical role of CCK in neuroplasticity of the motor cortex and motor skill learning. The CCKBRs of $Cck^{-/-}$ mice were not influenced by knocking out the $Cck$ gene, making it possible that the exogenous CCK activates the CCKBRs (*Feng et al., 2021*). CCK4, a tetrapeptide, can pass through the blood–brain barrier. $Cck^{-/-}$ mice with the defective motor learning capability improved significantly after the daily, single intraperitoneal injection of CCK4, to a comparable level as their WT control at Day 5. The results of the rescuing experiment imply a potential new target for facilitating motor rehabilitation.

# Materials and methods

## Animal

Young adult WT (C57BL/6) mice and C57BL/6 background transgenic mice, *Cck-Cre* (*Cck-ires-Cre*, Stock #012706, Jackson Laboratory) and $Cck^{-/-}$ (*Cck-CreER*, strain #012710, Jackson Laboratory), were used for behavior, electrophysiology and anatomy experiments. All mice were housed in the pathogen-free 12 hr light/dark cycle holding room with the temperature at 20–24°C. All experimental procedures were approved by the Animal Subjects Ethics SubCommittee of the City University of Hong Kong under the Animals (Control of Experiments) Ordinance (Cap. 340) (Licence number (21-206) in DH/HT&A/8/2/5 Pt.7).

## Single pellet reaching task

The behavioral experiment, single pellet reaching task, was modified based on a previously established procedure (*Xu et al., 2009*; *Chen et al., 2014*). A clear and transparent Plexiglas chamber (5 mm thickness, dimensions 20 cm × 15 cm × 8.5 cm) was built for mice training, with three 5 mm wide slits on the front wall: one is in the middle, the other two are 1.9 cm to the side, respectively. A 1.0-cm-height exterior shelf was affixed in front of the front wall to hold the chocolate pellets (#1811223, 20 mg, TestDiet) for reward. The food pellet was placed 0.7 cm away from the front wall

and 0.4 cm away from the midline of the slit, to encourage the mouse to use the dominant hand for catching (*Figure 1A*). The task has two periods, shaping and training. Mice were food restricted to keep approximately 90% body weight of the original weight during the whole process (*Figure 1B*). On shaping Day 1, two mice from the same cage were placed into the chamber for 20 min to acclimate to the environment; on shaping Day 2, an individual mouse was placed into the chamber for 20 min. During shaping, 10 food pellets were fed for each mouse every day to train mice eating food pellets. On shaping Day 3, a food tray full of food pellets was placed in front of the middle slit. The mouse can get the food reward by catching it through the slit with either hand. The experiment stopped when 20 times of reaching attempts were finished for each mouse. The dominant hand should be the one that shows over 70% preference. During the training period, mice reached for food pellets through the slit by the dominant hand, 40 attempts within 20 min every day. Only attempts by the dominant hand were counted. Based on the results of the attempts, the reaching attempts show four types: miss, no-grasp, drop, and success. A 'miss' means that the hand does not touch the food pellet. A 'no-grasp' means that the hand of the mouse touches the food pellet, but it does not successfully grasp the pellet. A 'drop' represents the mouse grasps the pellet, but it dropped due to whatever reasons during the retrieval. A 'success' was a reach in which the mouse successfully retrieved the pellet and put it into the mouth of the mouse. A high-speed camera was placed on the side of the chamber to videotape the reaching behavior of mice at 60 frames per second. The success rate was calculated as the number of successful attempts/the total attempts. The miss rate, the no-grasp rate, and the drop rate were also calculated to evaluate the performance of each step of mice. Hausdorff distances, the greatest of all the distances from a point in one set to the closest point in the other set, were calculated to assess the variation of trajectories.

## CCKBR antagonist injection

C57BL/6 mice were implanted a cannula in the motor cortex (coordinates: AP, 1.4 mm, ML, −/+1.6 mm, DV, 0.2 mm) contralateral to the dominant hand of the mice, followed by 3 days of recovery. Mice were grouped into Antagonist and Vehicle groups. L365.260 (CCKBR antagonist) (1 μl, 20 μM, Cat. No. 2767, biotechne) or vehicle (0.1% DMSO dissolved in ACSF) was injected into the motor cortex through the cannula with the flow rate of 100 nl/min pumped by a syringe pump (Hamilton, USA), before the mice were placed into the chamber for the single pellet reaching task training.

## Chemogenetic manipulation

A chemogenetics experiment was performed on *Cck-ires-Cre* mice (#012706, Jackson Laboratories). A Cre-dependent hM4Di virus was injected into the rhinal cortex. Detailed coordinates and volumes were described in the virus injection part. Besides, to specifically inhibit CCK-positive projections from the RC to the MC, a retro-Cre virus (AAV(retro)-EF1a-Cre-EGFP) was injected into the MC bilaterally and a Cre-dependent hM4Di virus (AAV-hSyn-DIO-hM4Di-mCherry) was injected into the RC of C57BL/6 mice. Mice were used for single pellet reaching task training 4 weeks post virus injection. Thirty minutes before behavior training, clozapine (0.4 mg/kg, Sigma-Aldrich, dissolved with 0.1% DMSO) was intraperitoneally injected to inactivate the activity of the CCK-positive neurons in the rhinal cortex. The same volume of vehicle (0.9% saline solution with 0.1% DMSO) was injected into mice injected with Cre-dependent hM4Di as a sham control group. A negative control virus (AAV8-hSyn-DIO-mCherry) combined with intraperitoneal clozapine injection was also carried out to exclude the influence of clozapine on motor learning ability.

## Virus injection and surgical process

AAV-hSyn-DIO-mCherry and AAV8-hSyn-DIO-hM4Di-mCherry were diluted to the titer around $5 \times 10^{12}$ copies/ml and AAV(retro)-EF1a-DIO-EYFP, AAV(retro)-EF1a-Cre-EGFP, AAV(retro)-hSyn-mCherry, AAV(retro)-EF1a-DO-mCherry-DIO-EGFP, and AAV-hSyn-CaMKII-GCaMP6s-SV40 were diluted to the titer around $1 \times 10^{13}$ copies/ml and injected into the mouse cortex as previously described (*Wu et al., 2014*; *Zhu and Roth, 2014*; *Tervo et al., 2016*). The mice were anesthetized with pentobarbital, and their fur between two ears trimmed and they were fixed on a stereotactic apparatus (RWD, China). Firstly, the head skin of the mouse was cleaned and sterilized with 70% alcohol and open to fully expose the skull. To accurately locate the regions of interest, the head was adjusted between the middle and the lateral, and the anterior and the posterior. To completely inactivate the rhinal

cortex, two injection sites per hemisphere were determined for virus injection using the following coordinates: site 1, anteroposterior (AP), −3.52 mm from Bregma, mediolateral (ML), 3.57 mm, dorso-ventral (DV), −3.33 mm from the brain surface; site 2, AP, −4.24 mm from Bregma, ML, 3.55 mm, DV, −2.85 mm from the brain surface. Additionally, to specifically inhibit the projections from the RC to the MC, retro-Cre virus was injected using the coordinates as: site 1, AP, 0.5 mm, ML, 1.35 mm, DV, 0.3 mm; site 2, AP, 1.2 mm, ML, 1.5 mm, DV, 0.3 mm; site 3, AP, 1.2 mm, ML, 1.5 ml, DV, 1.0 mm; site 4, AP, 1.8 mm, ML, 1.3 mm, DV, 0.3 mm. Microinjections were carried out using a microinjector (World Precision Instruments, USA) and a glass pipette (Cat#504949, World Precision Instruments, USA). The volume is 200 nl for each site and the flow rate is 50 nl/min.

To track the projection of CCK neuron from the rhinal cortex to the motor cortex, retrograde viruses, AAV-EF1a-DIO-eYFP, AAV-EF1a-DO-mCherry-DIO-EGFP, and AAV-hSyn-mCherry, was injected into the motor cortex. The coordinates is: site 1: AP, 1.8 mm to the Bregma, ML, 1.2 mm, DV, 0.2 and 0.6 mm; site 2: AP, 1.0 mm to the Bregma, ML, 1.5 mm, DV, 0.2 and 0.6 mm. The volume of each site at each DV was 200 nl and the flow rate was 20 nl/min to protect the fluid from flowing out. An anterograde AAV-hSyn-DIO-mCherry is also used for projection tracking by injecting the virus into the rhinal cortex of the hemisphere. The specific coordinates are as that for chemogenetic virus injection. After virus injection, skins were seamed with sterilized sutures, and an antibiotic ointment was applied to the incision to prevent infection and promote healing.

The surface virus infusion process for the calcium imaging was performed as described previously with slight modification (*Li et al., 2017*). A wide-tip glass pipette was prepared by a micropipette puller and then cut, polished, and flame treated to make it even and smooth. Mice were intraperi-toneally injected with dexamethasone (0.2 mg/kg, s.c.) and carprofen (5 mg/kg, s.c.) to protect the brain from swelling and inflammation. Three hours later, mice were anesthetized with pentobarbital. The periosteum covered on the skull was removed, cleaned, and dried with 100% alcohol to prevent the skull and tissues from growing. A $3 \times 3$ mm$^2$ window above the motor cortex contralateral to the dominant hand was opened with a hand drill, and the bone debris was carefully removed with fine forceps. After that, the dura around the injection area was removed (open a dura hole of about 1 mm$^2$) to expose the pial tissue for virus infusion. The tip of the pipette tightly covered the brain surface by lowering 400–500 μm, and 0.6 μl virus was infused at the speed of 0.06 μl/min. A $3 \times 3$ cover glass (thickness, around 150 μm) was attached to the brain surface, and gentle pressure was applied to keep the cover glass at the level same as the skull. The edge of the cover glass was sealed with superglue. After the glue totally hastened, the skin was stretched back and sutured.

## Baseplate implantation

Two to three weeks after cranial window implantation, the scalp over the skull was totally removed with surgical scissors. Success implantation shows a clear observation window without blood on the brain surface and a cover glass tightly fixed on the skull. The cover glass surface was gently cleaned with Ringer's solution and lens paper, and the regrowth of periosteum on the skull was removed with fine forceps. Before baseplate implantation, the skull was dried with 100% alcohol, covered with Metbond glue, and a thick layer of dental acrylic except for the cover glass for observation.

A one-photon miniscope (UCLA miniscope V4, Lab maker, Germany) connected to the data acqui-sition software was attached to the baseplate, secured on the stereotaxic micromanipulator, and grad-ually lowered to the cover glass until there was only a 1-mm gap between the skull and the baseplate. We turned on the LED and adjusted the focal distance of the electrowetting lens to 0 on the software. The position of the miniscope was adjusted until the brain tissue was observed in the data acquisition system. Dental acrylic was used to fix the baseplate to the acrylic cap covering the skull around the window. Once the dental acrylic had hardened, the miniscope was removed, and a metal cap was attached to the baseplate to protect the cover glass window.

## Calcium imaging and fluorescent signal analysis

After the implantation of the baseplate, a miniscope model was attached to the baseplate, and the mouse was placed in the chamber to acclimate to the weight of the miniscope for 20 min for 3 days. The LED laser and focal plate were slightly adjusted until the cells with fluorescent protruded from the background. A web camera was also connected to the data acquisition software and recorded the behavior movement of the animal simultaneously.

An imaging field of about 1.0 × 1.0 mm$^2$ (resolution: 608 × 608 pixels) video at approximately 10 min long was recorded. To clearly figure out the role played by CCK in the neuroplasticity of the motor cortex from *Cck$^{-/-}$* mice, C57BL/6 mice as well as C57BL/6 mice that intraperitoneal injection of CCKBR antagonist, L365.260 (0.4 mg/kg, Cat. No. 2767, biotechne). Raw AVI videos were firstly spatially down-sample by twofolds to reduce the size of the videos by Fiji (ImageJ, USA). Then a MATLAB algorithm, NoRMCorre, was applied for piecewise rigid motion correction before data analysis. The calcium signals were extracted with the MATLAB code of Constrained Nonnegative Matrix Factorization for microEndoscopic (CNMF-E) (code availability: https://github.com/zhoupc/CNMF_E; *Zhou et al., 2020*; *Zhou et al., 2018*). The scaled fluorescent calcium signal overtime was extracted as C_raw. The raw data were then deconvolved. The activity higher than three times the standard deviation of baseline fluctuation is deemed as a calcium event which has been revealed to be associated with neuronal spiking activity, and the rising phase of which was searched and used for further neuronal activity analysis (*Peters et al., 2014*; *Wang et al., 2017*). Timestamps from both the behavior videos and the calcium imaging videos were aligned to find out the time window when the mouse performed the reaching task. Neuronal activity in the time window from 100 ms before reaching to 100 ms after retrieval was considered the activity related to the movements. Wilcoxon ranksum test was conducted between activity inside the time window and activity outside ($p < 0.05$) to exclude the neurons that activated indiscriminately or not correlated with the reaching task. Neurons with the average activity in the time window higher than the average outside the time window were considered movement-related neurons. The neurons were aligned based on each neuron's sorted time of peak event to visualize each and all the neuronal activity patterns during the reaching task. The recurrence of neuronal activity related to the movements was also elevated by pairwise comparison of the population neuronal activity between trials using the Pearson correlation coefficient.

## Immunohistochemistry

Four weeks after virus injection, mice were perfused with 50 ml cold phosphate-buffered saline (PBS) (1×) to remove the blood and 50 ml 4% paraformaldehyde (PFA) in PBS to fix the brain tissue. The skull was carefully opened, and the brain was removed from the skull and fixed by immersing it in 4% PFA at 4°C for 24 hr, then dehydrated in 30% sucrose PBS solution until it sank to the bottom. Brains were covered with OTC, freezing fixed, and sectioned to a thickness of 50 μm using a freezing microtome (Leica, German). Brain slices were preserved in an anti-freezing solution (25% glycerol and 30% ethylene glycol, in PBS) and stored in the −80°C refrigerator.

For immunostaining, the brain slices were washed three times using 1× PBS in a shaker and incubated in blocking solution (10% normal goat serum and 0.2% Triton X-100 in PBS) for more than 1.5 hr in a shaker and incubated with the primary antibody (Mouse anti-GAD67, Millipore; Mouse anti-CaMK2a, Abcam; Mouse anti-mCherry, Invitrogen; Rabbit anti-Cre, Invitrogen) in 0.2% Triton and 5% Goat serum in PBS at 4°C for 24–36 hr. Slices were washed with PBS four times before incubating with the second antibody (Alexa Fluor 594-conjugated goat anti-mouse, Alexa Fluor 594-conjugated goat anti-rabbit, Jackson immunity) diluted in 0.1% Triton PBS solution for 3 hr. Finally, slices were washed in 1× PBS four times, then incubated with 4′,6-diamidino-2-phenylindole (DAPI) (1 mg/ml) for 10 min, mounted on slides and sealed with mounting medium (70% glycerol in PBS). Slices were observed and imaged with a confocal laser-scanning microscope (Zeiss, German) using ×10 and ×20 air objectives or ×40 and ×60 oil immersion objectives.

## Brain slice electrophysiology

The slice electrophysiology experiment was carried out following the methods reported previously (*Chen et al., 2019*). In the experiments, 6–8 weeks old C57BL/6 or *Cck$^{-/-}$* mice were anesthetized with isoflurane in a small chamber. The mouse head was cut, and the brain was rapidly removed and put into an oxygenated (95% O$_2$–5% CO$_2$) ACSF cold bath containing 26 mM of NaHCO$_3$, 2 mM of CaCl$_2$, 1.25 mM of KH$_2$PO$_4$, 1.25 mM of MgSO$_4$, 124 mM of NaCl, 3 mM of KCl, and 10 mM of glucose, pH 7.35–7.45. The brain was sectioned from the middle line into two hemispheres. The portions with the brain areas of the motor cortex were trimmed and glued on the ice-cold stage of a vibrating tissue slicer (Leika VT1000S). Coronal sections of slices containing the motor cortex (300 μm thick) were trimmed and gently transferred into an ACSF containing chamber, which was put in a water bath at

28°C and oxygen blowing continuously. After 2 hr of recovery in the ACSF bath, the slice was applied for the following electrophysiological recording.

A commercial 4-slice 8 × 8 channels recording system (MED, Panasonic Alpha-Med Sciences) was applied to record the fEPSPs. The MED probe is composed of 64 microelectrodes; the distance between the two channels is 50 × 50 μm (MED-P515A, 64-channel, 8 × 8 pattern, 50 × 50 μm, inter-electrode distance 150 μm or MED-PG515A).

After recovery, the motor cortex slice was covered by the recording electrodes. A fine-mesh anchor (Warner Instruments, Harvard) was covered on the brain slice to stabilize it, and the probe chamber was perfused with fresh ACSF oxygenated with oxygen with a peristaltic pump (Minipuls 3, Gilson), and the water bath was kept at 32°C. After 20 min of recovery, one of the microelectrodes in the area of interest was selected as the stimulating electrode through an inverted camera (DP70, Olympus). The surface layer of the motor cortex was stimulated with constant current pulses at 0.1ms in duration at 0.017 Hz by the connected controlling software, data acquisition software (Mobius, Panasonic Alpha-Med Sciences). After the baseline recording, which was stimulated at the currency of that triggering around 50% of the saturating potential. For drug application, CCK4 (final concentration: 500 nM) or vehicle was injected into the electrode dishes. HFS (25 bursts at 120 Hz for each burst, at the highest intensity) was applied to the simulation probe. The electrophysiological data were extracted and analyzed with offline software, Mobius software. For quantification of the LTP data, the initial amplitudes of fEPSPs were normalized and expressed as percentage changes over the averaged baseline activity. The fEPSP was normalized based on the percentage of the baseline potential.

## Rescue of the motor learning ability of the $Cck^{-/-}$ mice with CCK4

CCK4, a tetrapeptide derived from the peptide of CCK was selected as a potential drug to rescue the motor learning defect caused by the lack of CCK, because CCK4 remains the function to activate the CCK receptor but has a much smaller molecule than CCK8s or CCK58, making it transmit through the brain–blood barrier easily and smoothly (*Javanmard et al., 1999*; *Eser et al., 2009*). Therefore, intraperitoneal injection of the CCK4 is a simple and easily available way to rescue CCK lack caused motor learning defects.

After shaping, $Cck^{-/-}$ mice were injected intraperitoneally with CCK4 (0.45 mg/kg, Cat# ab141328, Abcam, UK) or vehicle before training every day.

## Statistical analysis

Group data were shown as mean ± standard error of the mean unless otherwise stated. Statistical analyses, including *t*-test, paired *t*-tests, one-way RM ANOVA, and two-way mixed ANOVA, were conducted in SPSS 26 (IBM, Armonk, NY). Statistical significance was defined as $p < 0.05$ by default.

## Acknowledgements

We thank Eduardo Lau for administrative and technical assistance. We also thank the following charitable foundations for their generous support to JFH: Wong Chun Hong Endowed Chair Professorship, Charlie Lee Charitable Foundation.

This work was supported by Hong Kong Research Grants Council, General Research Fund (GRF, 11103220 M and 11101521 M); Hong Kong Research Grants Council, Collaborative Research Fund (C7048-16g); Innovation and Technology Fund (MRP/053/18 X, GHP_075_19GD); Health and Medical Research Fund (HMRF, 09203656); National Natural Science Foundation of China (NSFC, 31671102); Innovation Technology Commission of the Hong Kong SAR, China (Health@InnoHK).

## Additional information

### Funding

| Funder | Grant reference number | Author |
| --- | --- | --- |
| Hong Kong Research Grants Council | GRF11103220M | Jufang He |

| Funder | Grant reference number | Author |
|--------|------------------------|--------|
| Hong Kong Research Grants Council | GRF11101521M | Jufang He |
| Hong Kong Research Grants Council, Collaborative Research Fund | C1043-21GF | Jufang He |
| Innovation and Technology Fund | MRP/053/18X | Jufang He |
| Innovation and Technology Fund | GHP_075_19GD | Jufang He |
| Health and Medical Research Fund | HMRF09203656 | Jufang He |
| National Natural Science Foundation of China | NSFC31671102 | Jufang He |
| Innovation Technology Commission of the Hong Kong SAR, China | Health@InnoHK | Jufang He |

The funders had no role in study design, data collection, and interpretation, or the decision to submit the work for publication.

## Author contributions

Hao Li, Formal analysis, Investigation, Visualization, Methodology, Writing - original draft, Project administration, Writing - review and editing; Jingyu Feng, Investigation, Methodology, Project administration; Mengying Chen, Min Xin, Formal analysis, Investigation; Xi Chen, Software, Visualization, Methodology, Project administration; Wenhao Liu, Data curation; Liping Wang, Resources; Kuan Hong Wang, Formal analysis, Supervision, Methodology, Project administration, Writing - review and editing; Jufang He, Resources, Data curation, Formal analysis, Supervision, Funding acquisition, Methodology, Project administration, Writing - review and editing

## Author ORCIDs

Hao Li http://orcid.org/0000-0001-5933-1443
Xi Chen http://orcid.org/0000-0002-2144-6584
Kuan Hong Wang http://orcid.org/0000-0002-2249-5417
Jufang He http://orcid.org/0000-0002-4288-5957

## Ethics

All experimental procedures were approved by the Animal Subjects Ethics SubCommittee of the City University of Hong Kong under the Animals (Control of Experiments) Ordinance (Cap. 340) (Licence number (21-206) in DH/HT&A/8/2/5 Pt.7).

## Decision letter and Author response

Decision letter https://doi.org/10.7554/eLife.83897.sa1
Author response https://doi.org/10.7554/eLife.83897.sa2

# Additional files

## Supplementary files

• MDAR checklist

## Data availability

All data generated or analyzed during this study can be downloaded through the link: https://doi.org/10.5061/dryad.9ghx3ffms.

The following dataset was generated:

| Author(s) | Year | Dataset title | Dataset URL | Database and Identifier |
|---|---|---|---|---|
| Li H, Feng J, Chen M, Xin M, Chen X, Wang K, He J | 2023 | Cholecystokinin facilitates motor skill learning by modulating neuroplasticity in the motor cortex | https://doi.org/10.5061/dryad.9ghx3ffms | Dryad Digital Repository, 10.5061/dryad.9ghx3ffms |

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

# Appendix 1

## Appendix 1—key resources table

| Reagent type (species) or resource | Designation | Source or reference | Identifiers | Additional information |
|---|---|---|---|---|
| Antibody | Anti-CamKIIa (Mouse monoclonal) | Abcam | Cat# Ab22609 | 1:500 |
| Antibody | Anti-GAD67 (Mouse monoclonal) | Millipore | Cat# MAB5406 | 1:1000 |
| Antibody | Anti-mCherry (Rabbit monoclonal) | Invitrogen | Cat# M11217 | 1:1000 |
| Antibody | Anti-mouse IgG Alexa 594 (Goat polyclonal) | Jackson ImmunoResearch | Cat# 115-585-003 | 1:500 |
| Antibody | Anti-rabbit IgG Alexa 594 (Goat polyclonal) | Jackson ImmunoResearch | Cat# 111-585-003 | 1:500 |
| Recombinant DNA reagent | AAV-hSyn-DIO-mCherry | Addgene | RRID: Addgene_50459 | |
| Recombinant DNA reagent | AAV(retro)-EF1a-Cre-EGFP | WZ Biosciences Inc | NA | |
| Recombinant DNA reagent | AAV-hSyn-DIO-hM4Di-mCherry | WZ Biosciences Inc | NA | |
| Recombinant DNA reagent | AAV-EF1a-DO-mCherry-DIO-EGFP | Braincase | Cat# BC0658 | |
| Recombinant DNA reagent | AAV-hSyn-DIO-hM4Di-mCherry | Addgene | RRID: Addgene_44362 | |
| Recombinant DNA reagent | retroAAV-EF1a-DIO-EYFP | Addgene | RRID: Addgene_27056 | |
| Recombinant DNA reagent | AAV-CamKIIa-GCaMP6s-WPRE-SV40 | Addgene | RRID: Addgene_107790 | |
| Chemical compound, drug | Pentobarbital | Alfasan International B.V. | NA | |
| Chemical compound, drug | Carprofen | Sigma-Aldrich | Cat# PHR1452 | |
| Chemical compound, drug | CCK-4 | Abcam, Cambridge, UK | Cat# ab141328 | |
| Chemical compound, drug | Dexamethasone | Sigma-Aldrich | Cat# D4902 | |
| Chemical compound, drug | Clozapine | Sigma-Aldrich | Cat# C6305 | |
| Chemical compound, drug | DAPI | Santa Cruz Biotechnology | Cat# sc-3598 | |
| Chemical compound, drug | Food pellet | TestDiet | Cat# 1811223 | |
| Genetic reagent (*Mus musculus*) | Mouse: C57BL/6 | The Laboratory Animal Services Centre, Chinese University of Hong Kong | NA | |
| Genetic reagent (*Mus musculus*) | Mouse: C57BL/6 | Laboratory Animal Research Unit, City University of Hong Kong | NA | |
| Genetic reagent (*Mus musculus*) | Mouse: Cck-ires-Cre | Jackson Laboratories | Stock# 012706 | |
| Genetic reagent (*Mus musculus*) | Mouse: Cck-CreER | Jackson Laboratories | Stock# 012710 | |
| Software, algorithm | Excel | Microsoft | https://www.microsoft.com/en-us/microsoft365/excel | |
| Software, algorithm | Matlab R2020a | Mathworks | https://www.mathworks.com/products/new_products/release2020a.html | |
| Software, algorithm | Fiji | *Schindelin et al., 2012* | https://imagej.net/Fiji | |
| Software, algorithm | Photoshop | Adobe | https://www.adobe.com/products/photoshop.html | |

*Appendix 1 Continued on next page*

*Appendix 1 Continued*

| Reagent type (species) or resource | Designation | Source or reference | Identifiers | Additional information |
|---|---|---|---|---|
| Software, algorithm | SPSS | IBM | https://www.ibm.com/products/spss-statistics; | |

