## [Editor Report]

This important study investigates the contribution of Cholecystokinin (CCK), a neurotransmitter known to be involved in sensory and emotional function, to motor learning. The authors provide convincing evidence combining behavioural assays, brain recording and stimulation experiments, knock out models, and targeted manipulations of several relevant pathways that support a contribution of rhinal CCK projections to motor cortex during learning in mice. This paper thus identifies a potential novel pathway involved in motor learning, but the specific contribution of rhinal CCK still needs to be fully characterised in future work given the extensive projections of rhinal CCK neurons to brain areas other than motor cortex.

---

## [Decision Letter]

**Decision letter after peer review:**

Thank you for submitting your article "Cholecystokinin from the Rhinal Cortex Facilitates Motor Skill Learning" for consideration by *eLife*. Your article has been reviewed by 3 peer reviewers, and the evaluation has been overseen by a Reviewing Editor and Tamar Makin as the Senior Editor. The following individual involved in the review of your submission has agreed to reveal their identity: Xulu Sun (Reviewer #2).

Essential revisions (for the authors):

Below follow the main points that should be addressed by the authors.

1) The authors should establish/discuss whether it can be concluded that the behavioral consequence of their various CCK manipulations is learning-specific.

2) The paper implicates motor cortex-projecting CCK neurons in the rhinal cortex as being a key component in motor learning. However, the relative importance of this pathway in motor learning is not pinned down, especially in terms of sufficiency and specificity. The Public Reviews have several specific comments about this, including adding new analyses and controls.

3) Related to the above: normal neural plasticity should be essential to motor skill learning throughout development not just during the current task. Did these CCK knock-out mice present any motor deficits that could have resulted from a lack of CCK-mediated neural plasticity during development?

4) The CCK4 rescue experiment demonstrates the sufficiency of CCK in promoting motor learning. However, it lacked specificity: IP injection did not allow specific "gain of function" in the motor cortex but instead, the improved learning ability in CCK knock-out mice could be a result of a global effect of CCK4 across multiple brain regions.

5) The Discussion has statements that are only weakly supported by the results (for example, lines 429-431, lines 432-433, and lines 447-448).

*Reviewer #1 (Recommendations for the authors):*

The paper is generally written and presented well, and the conclusions are interesting in principle, though it would substantially strengthen the paper to provide evidence for specificity both in the learning and in the CCK rhinal-motor cortex pathway. For example, in the experiments injecting CCK receptor antagonists, injections into the ipsilateral motor cortex would evidence area specificity, and injections after learning would demonstrate learning specificity. Likewise, in the experiments inhibiting rhinal CCK neurons, inhibiting specifically the motor cortex-projecting neurons would show pathway specificity, and performing this manipulation after learning would show learning specificity.

*Reviewer #2 (Recommendations for the authors):*

1) Based on the literature, CCK is widely distributed across multiple brain regions. I understand that the authors here focus on the motor system, but providing a more comprehensive behavioral assessment of CCK knock-out mice could be very useful to strengthen the claim that the impaired performance in the pellet-reaching task was not due to weakened overall learning ability (e.g., memory deficits) or pre-existing motor deficits but specific to learning new motor skills. It could also help readers have a better understanding of how CCK knock-out mice behave similarly or differently in the current study compared to mice injected with CCK antagonists only in the motor cortex, as well as compared to knock-out mice in previous studies.

2) To corroborate the claim that the observed motor learning deficits are specifically related to the rhinal-motor cortical projections, I'd suggest the following chemogenetic experiments: express hM4Di in rhinal cortical neurons and inject clozapine in the motor cortex to specifically inhibit the axon terminals of rhinal neurons there. Alternatively, if hM4Di cannot be sufficiently expressed in axon terminals, maybe express inhibitory opsins in rhinal cortical neurons and implant optic fibers in the motor cortex to inhibit the axon terminals of rhinal cortical projections. Then observe if animals with targeted inhibition of the rhinal-motor cortical projection also express motor learning deficits.

3) Histology: similar to my point in 1), I think providing a more complete picture of anatomical projections of CCK neurons would be helpful for comparing the current study to the literature. Therefore, I was wondering: in addition to Figure 4, could the authors show 3.1) the major downstream targets of rhinal CCK neural projections, and 3.2) the major upstream CCK-expressing brain regions that project to the motor cortex?

4) To hopefully help enhance the specificity of the rescue experiments (corresponding to my comments in public review point 5), I'd suggest that the authors inject CCK4 into the motor cortex of 4.1) CCK knock-out mice, and 4.2) mice with hM4Di-mediated rhinal cortical inhibition.

[Editors' note: further revisions were suggested prior to acceptance, as described below.]

Thank you for resubmitting your work entitled "Cholecystokinin from the Rhinal Cortex Facilitates Motor Skill Learning" for further consideration by *eLife*. Your revised article has been evaluated by Tamar Makin (Senior Editor) and a Reviewing Editor.

The manuscript has been improved but there are some remaining issues that need to be addressed, as outlined below:

1) The specific role of CCK from the RC to MC is still not entirely demonstrated. The activity of the RC-MC pathway is causally tested, and these neurons are shown to be almost entirely CCK positive, but the CCK antagonist and rescue manipulations are either systemic or MC-localized. It is therefore not directly shown that CCK in RC-MC neurons is the critical factor for learning. The paper is still interesting, although since this claim is the title of the paper, it may be worth reconsidering the title.

2) Another concern about the CCK antagonist experiment is that there might be CCK-positive, GABAergic neurons locally in the motor cortex, in addition to the axons projected from the rhinal CCK neurons. Injecting a CCK antagonist would thus likely lead to mixed effects from inhibiting local inhibitory neurons (i.e., disinhibition) + suppressing excitatory input from the rhinal cortex. I understand that this might be beyond the scope of the current study, but it would be important if the authors could point out this limitation when discussing their CCK antagonist results in the main text.

3) The major deficit in KO mice appears to be decreased variability and reach distance (Figure 1D), which results in increased misses (Figure 1G). This is mentioned in the discussion (lines 568-9), but not stated or quantified in the main text. If this is the case, it seems that this is an important point to make, since it gives clues about specific behavioral deficits that may be the source of impaired learning.

4) There seems to be missing figure panels in Figure 2 supplement 1. Both the main text and the figure legend referenced panels D and E, but I didn't see them in the actual figure. Related, I was wondering if the results described in lines 225 – 229 were from Antagonist vs. Vehicle mice rather than CCK-KO vs. WT mice; if so, please add panels D and E to Figure 2 and fix the relevant text.

5) The statistical tests in Figure 3H were done within each group and aimed to show whether there was any cross-day increase in neural activity correlation. However, it seems that if we compare the WT and antagonist groups, their day 6 neural activity correlations would be around the same level. Could the authors explain why? Was it because the antagonist group repetitively executed the same wrong movements (which could also be controlled by highly correlated, although "incorrect" neural activity)? If that's the case, the neural activity correlation score might not be an applicable indicator of motor learning in this context. If the authors were trying to show the neural plasticity underlying learning, then one metric that might be useful to measure is the correlation between neural activity on day 1 vs. day 6 (low cross-day correlations may suggest neural plasticity), assuming that the same neurons could be tracked across days.

6) The authors have provided evidence that restoring CCK in knock-out mice rescues motor function. However, they do not provide direct evidence that CCK enables motor cortical plasticity that causes this learning to happen -as claimed in the last sentence of the abstract, which should be edited. Showing this would require, for example, doing calcium imaging in the CCK-KO CCK4-injected mice and showing that motor cortical plasticity is also rescued in vivo. While we are not asking the authors to perform these additional experiments, they should discuss whether motor cortex is the only region responsible for the observed deficits, as well as how other areas not studied here could be confounding their results.

---

## [Author Response]

Essential revisions (for the authors):Below follow the main points that should be addressed by the authors.1) The authors should establish/discuss whether it can be concluded that the behavioral consequence of their various CCK manipulations is learning-specific.

Thanks for your comments and suggestions.

Movement ability of WT and *Cck^-/-^*

We test the stride length, stride time, step cycle ratio and the grasp force of both *Cck^-/-^* and WT mice.The result showed that the performance of *Cck^-/-^* and WT mice was comparable in terms of these four parameters, indicating that knockout of *cck* gene did not affect the basic movement ability (Figure1-figure supplement 1C, D, E, F). Besides, as we mentioned in the manuscript, the performance of *Cck^-/-^* and WT was similar at training day one, suggesting no damage to the motor performance of *Cck^-/-^* mice for single pellet reaching task.

Effect of antagonist on the performance of learned task

C57BL/6 mice learned the task were injected with the antagonist before training to test whether the antagonist caused any damage to the movement ability. The result showed that the performance had no significant change in terms of miss rate, no-grasp rate, drop rate or success rate after antagonist injection, indicating that the antagonist did not affect the movement ability of the mice (Figure2-figure supplement 1D).

Effect of chemogenetics manipulation on learned task

C57BL/6 mice injected with retro-Cre virus in the motor cortex together with the Cre-dependent hM4Di virus in the rhinal cortex were trained to learn the task. After learned the task, mice were injected with clozapine 30 minutes before training every day. Comparison of "before" and "after" clozapine injection, no significant difference were observed in terms of the percentage of "Miss", "No-grasp", "Drop" and "Success", suggesting that chemogenetics manipulation had no effect on movement ability (Figure 6-figure supplement 3D).

2) The paper implicates motor cortex-projecting CCK neurons in the rhinal cortex as being a key component in motor learning. However, the relative importance of this pathway in motor learning is not pinned down, especially in terms of sufficiency and specificity. The Public Reviews have several specific comments about this, including adding new analyses and controls.

Thanks for your comments and suggestions.

The specific of the projections from the rhinal cortex to the motor cortex for motor skill learning was tested. We first determined that over 98% of neurons in the rhinal cortex that projected to the motor cortex are CCK positive (Figure 6A, Figure 6-figure supplement 2A, D). Next, we injected the retro-Cre virus in the motor cortex and the Cre-dependent hM4Di in the rhinal cortex in C57BL/6 mice to specifically inhibit the CCK neurons from the rhinal cortex to the motor cortex. Compared to two control groups, the learning ability of the experimental group was significantly suppressed, suggesting that CCK projections from the rhinal cortex to the motor cortex are critical for motor skill learning (Figure 6). Detailed description was added in the part of "Result" in the manuscript.

3) Related to the above: normal neural plasticity should be essential to motor skill learning throughout development not just during the current task. Did these CCK knock-out mice present any motor deficits that could have resulted from a lack of CCK-mediated neural plasticity during development?

Thanks for your comments and suggestions.

Development is mainly gene-guided which prepares the physical structure for learning, while learning is dependent on the neural plasticity and a period of experience (such as motor training in this research). Besides, development is deemed as "experience-expectant", using common environmental information, while learning is "experience-dependent", sensitive to the specific individual experiences (Greenough et al., 1987; Galván, 2010). Moreover, development costs longer time to form a specific ability of a species in general. The role played by CCK in the development is not clear. Duchemin et al. (1987) studied the *cck* gene expression level in the brain of rats pre- and postnatally. They found that the *cck* mRNA was detectable on embryonic day 14 (E14) and gradually increased to the maximum level on postnatal day 14 (P14), indicating that CCK might participate in the development of rats. Paolo et al. (2007) mapped the expression of CCK in the mouse brain. Plentiful CCK expression was observed at E12.5 in the thalamus and spinal cord, and by E17.5, CCK expression extended to the cortex, hippocampus and hypothalamus, suggesting that CCK might also regulate the development of mice. Paolo et al. (2004) found that CCK suppressed the migration of GnRH-1 through CCK-A receptor in the brain. Besides, postnatal early learning may participate in development. CCK-B receptor antagonist administration (postnatal 6 hours) suppressed lambs get mother preference, indicating that CCK might be important for the development of mother preference of sheep. However, what the role played by CCK in the development of motor system is not well studied.

In this study, the performance of both *Cck^-/-^* and WT mice is at the same level without significant difference on Day one, in terms of the percentage of "miss", "no-grasp", "drop" and "success". Besides, the movement abilities, including stride length, stride time, step cycle ratio and grasp force, were comparable for both *Cck^-/-^* and WT mice (Figure 1-figure supplement 1C, D, E, F), suggesting that knockout of *cck* gene did not affect the basic movement ability. This could be because the development of basic movement ability is not learning-guided, but is physical structure-determined. We further applied CCK-BR antagonist and chemogenetic method to make up the potential defects of *Cck^-/-^* mice.

4) The CCK4 rescue experiment demonstrates the sufficiency of CCK in promoting motor learning. However, it lacked specificity: IP injection did not allow specific "gain of function" in the motor cortex but instead, the improved learning ability in CCK knock-out mice could be a result of a global effect of CCK4 across multiple brain regions.

Thanks for your suggestions.

First, the specificity of the circuit were studied by injecting a Cre virus in the MC and a Cre-dependent hM4Di virus in the RC. After injection with clozapine, the motor learning ability were significantly suppressed compared with the saline control and the control virus combined with clozapine.

Besides, we emphasized that the importance of the motor cortex in motor learning, not meant that other brain areas where also receive CCK-positive neuronal projections from the rhinal cortex, for example hippocampus (spatial memory), are not important for the performance of this task. Specific infusion the drug into the motor cortex is hard to rescue the motor learning ability of *Cck^-/-^* mice because the motor cortex is very large, varying from AP: -1.3 to 2.46 mm and ML: ±0.5 to ±2.75 mm and other areas receiving CCK projections from the rhinal cortex also could be important for motor learning. Direct infusion the drug into the motor cortex through a drug cannula, is hard to compensate the knock out of *cck* gene in the whole brain (Figure 7-figure supplement 1D, E). Besides, cannula implantation causes inescapable injury to the motor cortex, because the cannula must be inserted into the brain, so that the drug could be infused into the brain. This injury may affect the performance in the task, as the motor cortex is very critical for motor learning. Therefore, it is not the best method to be applied for motor skill learning ability rescuing.

Furthermore, CCK4 molecules can be transported to the whole brain by i.p. injection, as CCK4 is capable to pass through brain blood barrier, which compensates the knockout of *cck* gene in the whole brain, leading to the rescuing of motor learning ability. Furthermore, i.p. injection is widely accepted for drug discovery because it is very convenient, simply manipulated and does not cause any direct injury on the brain. Thus, we applied i.p. injection not only for whole brain CCK compensation, but also for the further study of the application in drug discovery.

5) The Discussion has statements that are only weakly supported by the results (for example, lines 429-431, lines 432-433, and lines 447-448).

The statements were revised.

Reviewer #1 (Recommendations for the authors):The paper is generally written and presented well, and the conclusions are interesting in principle, though it would substantially strengthen the paper to provide evidence for specificity both in the learning and in the CCK rhinal-motor cortex pathway. For example, in the experiments injecting CCK receptor antagonists, injections into the ipsilateral motor cortex would evidence area specificity, and injections after learning would demonstrate learning specificity. Likewise, in the experiments inhibiting rhinal CCK neurons, inhibiting specifically the motor cortex-projecting neurons would show pathway specificity, and performing this manipulation after learning would show learning specificity.

Thanks for your comments and suggestions.

In this study, we focus on the role played by CCK from the rhinal cortex, and how CCK affects motor learning. The single pellet reaching task was selected to study the role of CCK from the rhinal cortex in motor skill learning and the motor cortex is considered as the main area generates motor memory when training in this task (Komiyama et al., 2010; Peters et al., 2014; Richard et al., 2019). We emphasized that the importance of the contrallateral motor cortex in motor learning, not meant that other brain areas where also receive CCK-positive neural projections from the rhina cortex, for example hippocampus (spatial memory), are not important for the performance of this task. In fact, specifically inhibiting the projection from the rhinal cortex to the contrallateral motor cortex is not enough to suppress the motor learning ability, but inhibiting projecting in both sides of motor cortex (contro- and ipsi-lateral) could suppress the learning ability of mice, suggesting that the whole motor cortex is critical for motor skill learning (Figure 6, Figure 6-figure supplement 3). In this paper, we studied the relationship between the rhinal cortex and the motor cortex and the role played by CCK in this circuit. The specificity of the motor cortex is task-dependent, not the main purpose in this study.

The antagonist and chemogenetic manipulations were carried out after mice learned the skill of the task as you suggested, and the results suggested that antagonist or chemogenetic manipulation did not cause any defects to the movement ability of mice (Fgiure 2-figure supplement 1D, Figure 6-figure supplement 3D).

The specificity of the CCK-projection from the rhinal cortex to the motor cortex for motor skill learning was studied using chemogenetic methods in the revised version of the paper. We first determined that over 98% of neurons in the rhinal cortex that projected to the motor cortex are CCK positive (Figure 6A, Figure 6-figure supplement 2A, B). Next, we injected the retro-Cre virus in the motor cortex and the Cre-dependent hM4Di in the rhinal cortex in C57BL/6 mice to specifically inhibit the CCK neurons from the rhinal cortex to the motor cortex. Compared to two control groups, the learning ability of the experimental group was significant suppressed, suggesting that CCK projections from the rhinal cortex to the motor cortex are critical for motor skill learning (Figure 6). Detailed description was added in the part of "Result" in the manuscript.

Reviewer #2 (Recommendations for the authors):1) Based on the literature, CCK is widely distributed across multiple brain regions. I understand that the authors here focus on the motor system, but providing a more comprehensive behavioral assessment of CCK knock-out mice could be very useful to strengthen the claim that the impaired performance in the pellet-reaching task was not due to weakened overall learning ability (e.g., memory deficits) or pre-existing motor deficits but specific to learning new motor skills. It could also help readers have a better understanding of how CCK knock-out mice behave similarly or differently in the current study compared to mice injected with CCK antagonists only in the motor cortex, as well as compared to knock-out mice in previous studies.

Thanks for your comments.

We test the stride length, stride time, step cycle ratio and the grasp force of both *Cck^-/-^* and WT mice. The result showed that the performance of *Cck^-/-^* and WT mice was comparable in terms of these four parameters, indicating that knockout of *cck* gene did not affect the basic movement ability (Figure 1-figure supplement 1C, D, E, F). Besides, as we mentioned in the manuscript, the performance of*Cck^-/-^* and WT was similar at training day one, suggesting no damage to the motor performance of *Cck^-/-^* mice for single pellet reaching task.

We added the comparison of the performance of *Cck^-/-^* and the antagonist group in the “Discussion” part of the revised manuscript.

2) To corroborate the claim that the observed motor learning deficits are specifically related to the rhinal-motor cortical projections, I'd suggest the following chemogenetic experiments: express hM4Di in rhinal cortical neurons and inject clozapine in the motor cortex to specifically inhibit the axon terminals of rhinal neurons there. Alternatively, if hM4Di cannot be sufficiently expressed in axon terminals, maybe express inhibitory opsins in rhinal cortical neurons and implant optic fibers in the motor cortex to inhibit the axon terminals of rhinal cortical projections. Then observe if animals with targeted inhibition of the rhinal-motor cortical projection also express motor learning deficits.

Thanks for your comments and suggestions.

We added some experiments based on your suggestions. We injected a retrograde AAV- Cre virus into the motor cortex and a Cre dependent hM4Di virus into the rhinal cortex to specifically inhibit the CCK projection from the rhinal cortex to the motor cortex. The superficial neurons in the motor cortex is very important for motor skills learning, as it receives CCK projection from the rhinal cortex (Figure 4B). Drug cannula or optic fiber implantation will injury the superficial layer of the motor cortex, which may impair the motor learning ability of mice.

The specificity of the CCK-projection from the rhinal cortex to the motor cortex for motor skill learning was studies using chemogenetic methods in the revised version of the paper. We first determined that over 98% of neurons in the rhinal cortex that projected to the motor cortex are CCK positive (Figure 6A, Figure 6-figure supplement 2A, B). Next, we injected the retro-Cre virus in the motor cortex and the Cre-dependent hM4Di in the rhinal cortex in C57BL/6 mice to specifically inhibit the CCK neurons from the rhinal cortex to the motor cortex. Compared to two control groups, the learning ability of the experimental group was significant suppressed, suggesting that CCK projections from the rhinal cortex to the motor cortex are critical for motor skill learning (Figure 6). Detailed description was added in the part of "Result" in the manuscript.

3) Histology: similar to my point in 1), I think providing a more complete picture of anatomical projections of CCK neurons would be helpful for comparing the current study to the literature. Therefore, I was wondering: in addition to Figure 4, could the authors show 3.1) the major downstream targets of rhinal CCK neural projections, and 3.2) the major upstream CCK-expressing brain regions that project to the motor cortex?

Thanks for your comments and suggestions.

The motor cortex also receive CCK projections from other cortices, such as the contralateral motor cortex, the deep layer of visual cortex and auditory cortex, and thalamus (Figure 4-figure supplement 1A, B, C). The downstream targets of rhinal CCK projections, including the auditory cortex, visual cortex, hippocampus, and amygdala, were studied in our lab before (Li et al.,2014; Chen et al., 2019; Su et al. 2019; Feng et al. 2021). Other brain areas, such as the prefrontal cortex, lateral septal nucleus, agranular insular cortex, and accumbens nucleus, receiving CCK projections from the rhinal cortex were showed in the Figure 4-figure supplement 1D.

4) To hopefully help enhance the specificity of the rescue experiments (corresponding to my comments in public review point 5), I'd suggest that the authors inject CCK4 into the motor cortex of 4.1) CCK knock-out mice, and 4.2) mice with hM4Di-mediated rhinal cortical inhibition.

Thanks for your comments.

First, the specificity of the circuit were studied by injecting a Cre virus in the MC and a Cre-dependent hM4Di virus in the RC. After injection with clozapine, the motor learning ability were significantly suppressed compared with the saline control and the control virus combined with clozapine.

Besides, we emphasized that the importance of the motor cortex in motor learning, not meant that other brain areas where also receive CCK-positive neural projections from the rhinal cortex, for example hippocampus (spatial memory), are not important for the performance of this task. Specific infusion the drug into the motor cortex is hard to rescue the motor learning ability of *Cck^-/-^* mice because the motor cortex is very large, varying from AP: -1.3 to 2.46 mm and ML: ±0.5 to ±2.75 mm and other areas receiving CCK projections from the rhinal cortex also could be important for motor learning. Actually, we tried to inject CCK into the motor cortex through a drug cannula, but the result showed that it is hard to compensate the knock out of *cck* gene in the whole brain, and rescue the motor learning ability (Figure 7-figure supplement 1D, E). Moreover, cannula implantation causes inescapable injury to the motor cortex, because the cannula must be inserted into the brain, so that the drug could be infused into the brain. This injury may affect the performance in the task, as the motor cortex is very critical for motor learning. Therefore, it is not the best method to be applied for motor skill rescuing.

Furthermore, CCK4 molecules can be transported to the whole brain by i.p. injection, as CCK4 is capable to pass through brain blood barrier, which compensates the knockout of *cck* gene in the whole brain, leading to the rescuing of motor learning ability. Furthermore, i.p. injection is widely accepted for drug discovery because it is very convenient, simply manipulated and does not causes any direct injury on the brain. Thus, we applied i.p. injection not only for whole brain CCK compensation, but also for the further study of the application in drug discovery.

[Editors' note: further revisions were suggested prior to acceptance, as described below.]

The manuscript has been improved but there are some remaining issues that need to be addressed, as outlined below:1) The specific role of CCK from the RC to MC is still not entirely demonstrated. The activity of the RC-MC pathway is causally tested, and these neurons are shown to be almost entirely CCK positive, but the CCK antagonist and rescue manipulations are either systemic or MC-localized. It is therefore not directly shown that CCK in RC-MC neurons is the critical factor for learning. The paper is still interesting, although since this claim is the title of the paper, it may be worth reconsidering the title.

Thanks for your comments and suggestions. The title was revised to "Cholecystokinin facilitates motor skill learning by modulating neuroplasticity in the motor cortex".

2) Another concern about the CCK antagonist experiment is that there might be CCK-positive, GABAergic neurons locally in the motor cortex, in addition to the axons projected from the rhinal CCK neurons. Injecting a CCK antagonist would thus likely lead to mixed effects from inhibiting local inhibitory neurons (i.e., disinhibition) + suppressing excitatory input from the rhinal cortex. I understand that this might be beyond the scope of the current study, but it would be important if the authors could point out this limitation when discussing their CCK antagonist results in the main text.

Thanks for your suggestions. We added it to the Discussion in line 635-639.

3) The major deficit in KO mice appears to be decreased variability and reach distance (Figure 1D), which results in increased misses (Figure 1G). This is mentioned in the discussion (lines 568-9), but not stated or quantified in the main text. If this is the case, it seems that this is an important point to make, since it gives clues about specific behavioral deficits that may be the source of impaired learning.

Thanks for your comments and suggestions.

In our manuscript, Figure 1E shows that the variability of reach trajectories (Hausdorff Distance) is significantly reduced with training in WT mice, but not in *Cck^-/-^* mice. Following the reviewer’s comments, we also compared the reaching variability between WT and *Cck^-/-^* mice on Day1, but there is no significant difference (*Cck*^-/-^ vs. WT, 0.4997 ± 0.1080 cm vs. 0.5320 ± 0.1358 cm, p = 0.59). The behavioral variations between individual animals of the same genotype might require larger sample size to detect any potential significance in between-subject comparisons. On the other hand, what we reported in Figure 1E is based on within-subject comparison of learning-related reduction in motor variability, which is present in WT mice but impaired in *Cck^-/-^* mice. We have removed the inaccurate statements in the Discussion.

4) There seems to be missing figure panels in Figure 2 supplement 1. Both the main text and the figure legend referenced panels D and E, but I didn't see them in the actual figure. Related, I was wondering if the results described in lines 225 – 229 were from Antagonist vs. Vehicle mice rather than CCK-KO vs. WT mice; if so, please add panels D and E to Figure 2 and fix the relevant text.

Thanks for your comments and suggestions and we revised the mistake. There may be something wrong with the submitted Figure 2—figure supplement 1. We resubmitted this figure and hope you can find D and E in this figure.

5) The statistical tests in Figure 3H were done within each group and aimed to show whether there was any cross-day increase in neural activity correlation. However, it seems that if we compare the WT and antagonist groups, their day 6 neural activity correlations would be around the same level. Could the authors explain why? Was it because the antagonist group repetitively executed the same wrong movements (which could also be controlled by highly correlated, although "incorrect" neural activity)? If that's the case, the neural activity correlation score might not be an applicable indicator of motor learning in this context. If the authors were trying to show the neural plasticity underlying learning, then one metric that might be useful to measure is the correlation between neural activity on day 1 vs. day 6 (low cross-day correlations may suggest neural plasticity), assuming that the same neurons could be tracked across days.

Thanks for your comments and suggestions.

Layer 2/3 of motor cortex is considered as the main layer driving neurons in the deep layer to produce motor cortex output. The neuronal activity of layer 2/3 is correlated with the movement of mice and the correlation increases with motor training (Peters et al., 2014). We compared the trial-to-trial population activity correlation of Day 1 to that of Day 6 to demonstrate that the neuroplasticity of neurons in layer 2/3 of *Cck*^-/-^ and Antagonist group was significantly suppressed due to the lack of CCK. We deemed the neuronal activity correlation on Day 1 without prior motor training as the baseline, and the increase of correlation on Day 6 compared with that on Day 1 as an index of motor learning, similar to the previous study (Peters et al. 2014).

The baseline of neuronal activity correlation of *Cck*^-/-^ and Antagonist groups may be directly affected by CCK, as many CCK-positive neurons, including inhibitory and excitatory neurons, are located in the layer 2/3 of the motor cortex (Watakabe et al., 2012). Higher baseline of *Cck*^-/-^ and Antagonist may be because the fact that deficiency of CCK suppressed the exploration of the optimal path and abandonment of ineffective movements that would otherwise occur in wildtype mice. Because the neuroplasticity was suppressed in *Cck*^-/-^ and Antagonist groups, the neuronal correlation did not increase with training.

Besides, the correlation between trials is only one dimension for assessing the neuronal activity changes with learning. The “activated population activity” (Figure 3E, F) was also applied to show that CCK deficiency could suppress the neuronal plasticity of *Cck*^-/-^ and Antagonist groups.

The correlation of neuronal activity between Day 1 and Day 6 also can be used to assess the neuronal activity changes with learning. However, neurons detected by one-photon miniscope on Day 1 and Day 6 were not totally the same because motor training also shaped the pattern of activated neurons: some neurons were activated on Day 1, but quiescent on Day 6, vice versa. Therefore, it is hard to definitively track all the same neurons through the training period using the miniscope.

To address the limitation highlighted in the reviewer’s comments, we calculated the "increase of trial-to-trial population activity correlation" to make it more comprehensible for readers (Figure 3—figure supplement 2).

6) The authors have provided evidence that restoring CCK in knock-out mice rescues motor function. However, they do not provide direct evidence that CCK enables motor cortical plasticity that causes this learning to happen -as claimed in the last sentence of the abstract, which should be edited. Showing this would require, for example, doing calcium imaging in the CCK-KO CCK4-injected mice and showing that motor cortical plasticity is also rescued in vivo. While we are not asking the authors to perform these additional experiments, they should discuss whether motor cortex is the only region responsible for the observed deficits, as well as how other areas not studied here could be confounding their results.

Thanks for your comments and suggestions. We revised the statements in the Abstract and discussed the role of the motor cortex and other brain areas in the "Discussion".